# Self-wetting triphase photocatalysis for effective and selective removal of hydrophilic volatile organic compounds in air

Fei He[1], Seunghyun Weon [2], Woojung Jeon[1], Myoung Won Chung [2] & Wonyong Choi [1✉]

Photocatalytic air purification is widely regarded as a promising technology, but it calls for more efficient photocatalytic materials and systems. Here we report a strategy to introduce an in-situ water (self-wetting) layer on $WO_3$ by coating hygroscopic periodic acid (PA) to dramatically enhance the photocatalytic removal of hydrophilic volatile organic compounds (VOCs) in air. In ambient air, water vapor is condensed on $WO_3$ to make a unique tri-phasic (air/water/$WO_3$) system. The in-situ formed water layer selectively concentrates hydrophilic VOCs. PA plays the multiple roles as a water-layer inducer, a surface-complexing ligand enhancing visible light absorption, and a strong electron acceptor. Under visible light, the photogenerated electrons are rapidly scavenged by periodate to produce more •OH. PA/$WO_3$ exhibits excellent photo-catalytic activity for acetaldehyde degradation with an apparent quantum efficiency of 64.3% at 460 nm, which is the highest value ever reported. Other hydrophilic VOCs like formaldehyde that are readily dissolved into the in-situ water layer on $WO_3$ are also rapidly degraded, whereas hydrophobic VOCs remain intact during photocatalysis due to the "water barrier effect". PA/$WO_3$ successfully demonstrated an excellent capacity for degrading hydrophilic VOCs selectively in wide-range concentrations (0.5−700 ppmv).

[1] Division of Environmental Science and Engineering, Pohang University of Science and Technology (POSTECH), Pohang 37673, Korea. [2] School of Health and Environmental Science, Korea University, Seoul 02841, Korea. ✉email: wchoi@postech.edu

Volatile organic compounds (VOCs) are major components of air pollution, which significantly deteriorate air quality and seriously affect human health[1–3]. A common control method for VOCs is adsorption using porous medium (e.g., activated carbon, zeolite, MOFs, etc.)[4,5], but their equilibrium adsorption capacity decreases significantly with lowering VOCs concentration[6]. Photocatalysis is widely considered as a promising method for air purification because of its ability to operate under ambient temperature and pressure conditions and to degrade and mineralize VOCs[7–9]. Photocatalytic degradation (PCD) maintains its removal efficiency even in the low concentration range[3,10], which is more advantageous to deal with VOCs in sub-ppm levels (e.g., indoor air)[11]. Considering that visible light takes a much higher proportion (~43%) of solar light than UV light (~4%) and a dominant portion of indoor light, it is essential to develop visible-light-responsive photocatalysts for practical application of air purification[12]. However, the performance of visible-light-driven photocatalysts is generally much lower than UV photocatalysis and needs significant improvements to satisfy the requirements for practical air purification[13]. As the PCD of VOCs is initiated mainly by hydroxyl radical (•OH) attack[14–16], an efficient way of enhancing visible light PCD is to facilitate the generation of •OH.

WO$_3$ is one of the most frequently investigated photocatalysts with notable visible light activity ($E_g \approx 2.8$ eV)[10,17,18], which is also stable in oxidative and acidic condition[19]. Although its valence band (VB) edge potential (at about 3.0 $V_{NHE}$) is positive enough to generate OH radicals (•OH/H$_2$O, $E^0 = +2.8\ V_{NHE}$)[13], its conduction band (CB) edge potential (~0.4 $V_{NHE}$)[10] is not negative enough to make CB electrons scavenged by O$_2$ via a single-electron transfer (e.g., O$_2$/O$_2$•$^-$, $E^0 = -0.33\ V_{NHE}$; O$_2$/HO$_2$•, $E^0 = -0.05\ V_{NHE}$)[13,20]. As a result, the photocatalytic activity of WO$_3$ is highly limited due to the rapid charge recombination[21,22]. Many attempts have been made to increase the visible light activity of WO$_3$ and one of the most effective methods is to load Pt nanoparticles as a co-catalyst[23], which enables the multi-electron reduction of O$_2$ to H$_2$O$_2$ or H$_2$O at the WO$_3$ CB edge potential (O$_2$/H$_2$O$_2$, $E^0 = +0.68\ V_{NHE}$; O$_2$/H$_2$O, $E^0 = +1.23\ V_{NHE}$)[24–26] with facilitating the charge separation and subsequently OH radical production[27]. However, the use of expensive Pt cocatalyst limits its practical applications[28], which calls for a more economical method utilizing cheaper material. In typical PCD mechanisms working on the gas–solid interface, photogenerated holes most likely react with adsorbed water molecules (or surface hydroxyl groups) to form •OH[11,29]. Water molecules are not only a source of •OH but also an adsorbent competing with VOCs for the surface sites[30,31]; the dual effects of water vapor affect the PCD of VOCs differently depending on the hydrophilic and hydrophobic nature of VOCs[32].

In this work, we propose to utilize water in a uniquely different way for the removal of VOCs by introducing an in situ water layer on the photocatalyst surface in ambient air. The presence of a thin surface water layer selectively solubilizes and concentrates hydrophilic VOCs in it, which subsequently facilitates their PCD. To achieve this, periodic acid (PA, HIO$_4$·2H$_2$O) as a highly hygroscopic substance was employed to induce an in situ water layer formed between the photocatalyst surface and air phase; as a result, the gas–solid interface can be transformed into a gas–liquid–solid (triphase) interface. In addition, the highly oxidizing PA[33,34] has the potential to serve as a strong scavenger of CB electrons, which should enhance the formation of •OH via the hole transfer. Herein, the PA/WO$_3$ system was employed for the PCD of several VOCs to propose a concept of PA-assisted photocatalysis that employs the in situ formation of the water layer on the photocatalyst surface. The low-cost PA-loaded WO$_3$ exhibits even higher activity than Pt-loaded WO$_3$ for the PCDs of hydrophilic VOCs.

This presents a cost-effective technology for high-performance selective degradation of hydrophilic VOCs.

## Results

**Enhanced activities of PA-coated photocatalysts.** PA was combined with three common visible-light photocatalysts (BiVO$_4$, N-TiO$_2$, and WO$_3$), and their photocatalytic activities were tested for acetaldehyde (AA) degradation in a closed-circulation reactor (Supplementary Fig. 1) under visible light ($\lambda > 420$ nm). Each PCD test consisted of a dark circulation period (20 min) for adsorption equilibrium and the following irradiation period. Figure 1a shows the degradation of AA and the accompanying production of CO$_2$ over bare and PA-treated WO$_3$, N-TiO$_2$, and BiVO$_4$. The PCD rate constant ($k_d$), removal efficiency, and mineralization efficiency are summarized in Supplementary Table 1. The photocatalytic activities of WO$_3$, N-TiO$_2$, and BiVO$_4$ were significantly improved by the PA treatment among which PA/WO$_3$ exhibited the highest PCD activity. PA also has a similar PCD-promoting effect for TiO$_2$ (P25) under UV irradiation (Supplementary Fig. 2), which confirms that the PA effect is the same regardless of the kind of photocatalysts. Interestingly, the photocatalytic activities of PA/P25 and PA/WO$_3$ are similar under LED irradiation of 365 nm while PA/WO$_3$ is far more active than PA/P25 under 460 nm LED irradiation (see Supplementary Figs. 2 and 3). It is particularly notable that PA/WO$_3$ exhibited higher activity than Pt/WO$_3$ which is one of the most active visible-light photocatalysts[10,23,27]. The optimal composition of PA/WO$_3$ was tested by varying the mixing ratio of PA:WO$_3$ which exhibits good activity for a wide range of PA:WO$_3$ mass ratios between 2/3 and 3/2 (see Supplementary Fig. 4). The highest PCD activity was observed at the 1:1 mass ratio of PA:WO$_3$, which was used in the preparation of PA/WO$_3$. As shown in Fig. 1b and c, the AA degradation and the concurrent CO$_2$ production over PA/WO$_3$ were about 2.7 times and 4.0 times higher than that of Pt/WO$_3$, respectively. It should be also noted that PA/WO$_3$ exhibited similar enhanced PCD activities under different irradiation conditions of blue LED ($\lambda = 460$ nm), UV LED ($\lambda = 365$ nm) and halogen lamp ($\lambda > 420$ nm) (Supplementary Fig. 3, Supplementary Table 2). The AQE of PA/WO$_3$ in the PCD of AA reached 16.1% and 22.3% under blue LED ($\lambda = 460$ nm) and UV LED ($\lambda = 365$ nm) irradiation, respectively, which were much higher than that of bare WO$_3$ (Supplementary Fig. 5). PA/WO$_3$ rapidly degraded AA of different initial concentrations ($[AA]_0 = 120–700$ ppmv) under blue LED irradiation (Supplementary Fig. 6). As $[AA]_0$ increased, AQE continued to increase until $[AA]_0$ reached 600 ppmv. This indicates that more AA molecules are concentrated in the surface region with increasing the gas-phase AA concentration, implying the unique AA-accumulating behavior of PA/WO$_3$. An AQE as high as 64.3% was achieved when $[AA]_0$ was 700 ppmv. As far as we know, this AQE value is significantly higher than any other reported ones for the visible-light-driven PCD of AA (Supplementary Table 3). In addition, the AQE of PA/WO$_3$ is also remarkable to our knowledge among the reported WO$_3$-based visible-light PCD systems (Supplementary Table 4). To check the long-term durability of the PA component, the PA/WO$_3$ sample that had been stored for 6 months under ambient conditions was tested for its PCD activity, which was little different from that of the fresh PA/WO$_3$. This demonstrates that the PA/WO$_3$ sample can be kept in long-term storage without losing its catalytic activity (see Supplementary Fig. 7). The above results showed the excellent performance of PA/WO$_3$ as a visible-light photocatalyst for the degradation of AA that is a common indoor air pollutant.

To clarify the synergistic effect between PA and WO$_3$, other iodine-containing inorganic compounds including NaIO$_4$, NaIO$_3$, NaI, and HIO$_3$ were also tested for their effects on the PCD activity of WO$_3$. Their activity decreased in the following order:

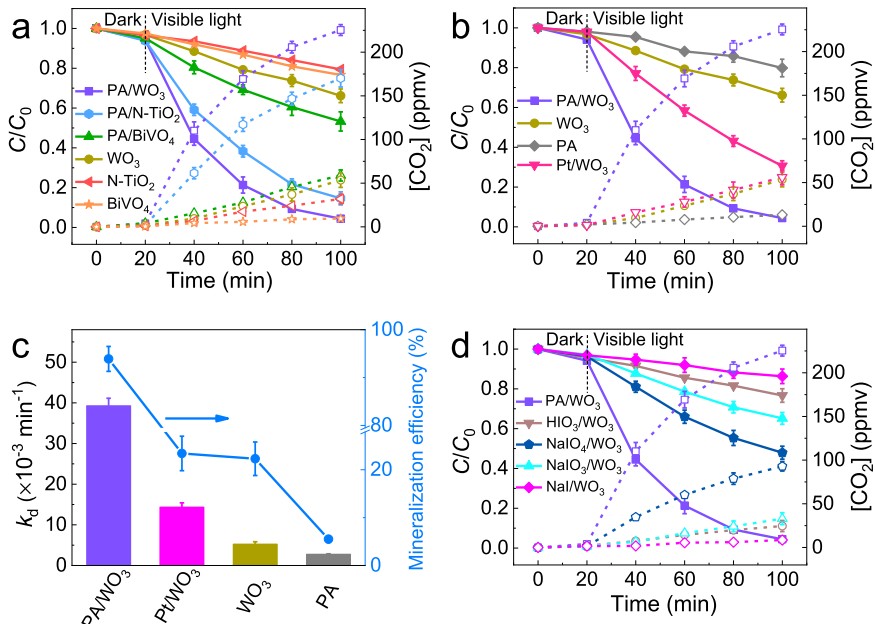

**Fig. 1 Visible light-driven PCD of acetaldehyde (AA). a** The time profiles of the PCD of AA and the accompanying production of $CO_2$ on bare and PA-treated $WO_3$, N-$TiO_2$, and $BiVO_4$. **b** The PCD comparison between PA/$WO_3$ and Pt/$WO_3$. **c** The PCD rate constant ($k_d$) and the mineralization efficiency after 80 min reaction for PA/$WO_3$, Pt/$WO_3$, bare PA, and $WO_3$. **d** PCD activities of $WO_3$ treated with different iodine reagents. The dashed lines with open symbols represent the $CO_2$ concentration generated from AA degradation. Error bars are defined as standard deviation. Experimental conditions: $[AA]_0 = 120$ ppmv; visible light ($\lambda > 420$ nm) intensity of 2.2 mW/cm$^2$; sample amount of 50 mg; RH 65%; reaction temperature of 30 °C.

PA/$WO_3$ » NaIO$_4$/$WO_3$ > NaIO$_3$/$WO_3$ ≈ $WO_3$ > HIO$_3$/$WO_3$ > NaI/$WO_3$ (Fig. 1d and Supplementary Table 1). Only periodate compounds showed obvious enhancement effects on the PCD activity of $WO_3$. In general, periodate-based compounds with high-valence-state iodine exhibit stronger electron-accepting and oxidizing capacity. The standard 2-electron reduction potential of IO$_4^-$ was much higher than that of IO$_3^-$ [$E^0$ (IO$_4^-$/IO$_3^-$) = +1.623 V$_{NHE}$][35,36] and ($E^0$ (IO$_3^-$/HIO$_2$) ≈ +0.88 V$_{NHE}$)[37]. Therefore, periodate ions should be responsible for the efficient trapping of CB electrons in the combined $WO_3$–IO$_4^-$ system. However, it is interesting to note that the PCD activity of NaIO$_4$/$WO_3$ ($k_d = 9.65 \times 10^{-3}$ min$^{-1}$) was much lower than that of PA/$WO_3$ ($k_d = 39.3 \times 10^{-3}$ min$^{-1}$). We observed that the surface of the PA/$WO_3$ catalyst became wet spontaneously after its exposure to ambient air while this phenomenon did not occur at all in the case of NaIO$_4$/$WO_3$. A similar effect of PA was also found on N-$TiO_2$ (Supplementary Fig. 8). Based on this observation, we hypothesized that this spontaneously formed surface layer may influence the photocatalytic activity of PA/$WO_3$ during the PCD of AA and further investigated the effect of water on the PCD activity.

The above PCD experiments employed high concentrations of AA ranging in 120–700 ppmv, which is unrealistically high for indoor environments. To demonstrate the performance of the PA/$WO_3$ photocatalyst in a more realistic condition, the PCD of formaldehyde (FA) on PA/$WO_3$ was additionally tested at a much lower concentration of 500 ppbv in a larger reactor (1.5 L) (compared with the PCD condition of AA) (see Fig. 2). FA is a common indoor air pollutant and a human carcinogen classified by the World Health Organization (WHO). The FA PCD tests were conducted under blue LED ($\lambda = 460$ nm) irradiation. After 30 min PCD reaction, the concentration of FA decreased from 500 to 58 ppbv (lower than the limit concentration allowed by WHO, 80 ppbv[38]) over 50 mg PA/$WO_3$. Note that FA could be removed by PCD using as low as 1 mg PA/$WO_3$. The activity of PA/$WO_3$ was far higher than bare $WO_3$ (Fig. 2b), and it remained active even after 10 PCD cycles (Fig. 2c). The above results

confirm that PA/$WO_3$ has an excellent capacity for degrading hydrophilic VOCs (FA and AA) selectively in wide-range concentrations (500 ppbv–700 ppmv). No other existing indoor air purification methods have such effective, durable, and selective capacity for the removal of indoor aldehydes under ambient conditions.

**Roles of PA and in situ formed water layer.** PA molecules with hydroxyl groups are highly hygroscopic due to the hydrogen bonding between water molecules and the hydroxyl groups in PA[39]. On the other hand, NaIO$_4$ is not hygroscopic due to the absence of hydroxyl groups. To evaluate the water absorption capacity, fresh air with 65% RH was flowed over the surface of bare $WO_3$, NaIO$_4$/$WO_3$, and PA/$WO_3$ samples for 30 min; then, the weight increased in each catalyst sample was measured (Fig. 3a). The weight of bare $WO_3$ and NaIO$_4$/$WO_3$ was not changed while that of PA/$WO_3$ increased by ~26%, which can be attributed to liquid water condensed on PA/$WO_3$ from the humid air. Although water molecules should be adsorbed on the bare $WO_3$ surface via hydrogen bonding with the surface hydroxyl groups[40], the adsorbed water molecules do not induce the condensation of the water layer on $WO_3$. It seems that the presence of PA facilitates the condensation of the water layer on the surface of the catalyst. To further confirm the in situ formation of the water layer via condensation, the PA/$WO_3$ coated on the glass was exposed in ambient air for 10 min, then the sample was in situ observed with microphotography (see the Supplementary video). We observed that the formation of condensing water layer induces the movement of $WO_3$ particles. When the wet PA/$WO_3$ sample was dried by an external heater, PA crystals appeared with ceasing the moving of $WO_3$ particles. Upon stopping the heating, the water absorption process resumed with redissolving PA crystals and moving $WO_3$ particles again. The FT-IR spectral changes in PA/$WO_3$ and bare $WO_3$ before and after the exposure to humid air also show such a trend clearly (Fig. 3b). The peak assigned to the stretching vibration of surface OH is located at

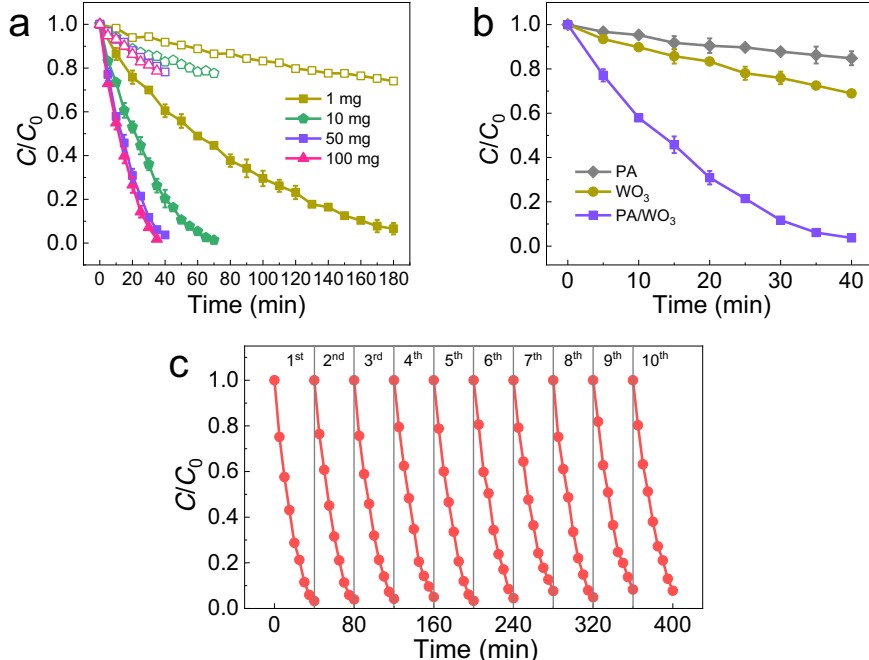

**Fig. 2 Visible light-driven PCD of formaldehyde (FA) at 500 ppbv on PA/WO₃.** **a** The dark control tests (open symbols) and the PCD tests (filled symbols) over PA/WO₃ (1:1) with different catalyst mass (e.g., 10 mg catalyst composed of 5 mg PA and 5 mg WO₃). **b** PCD of FA on PA/WO₃, bare WO₃ and PA. Error bars are defined as standard deviation. **c** Repeated PCD cycles of FA degradation over 50 mg PA/WO₃. Experimental conditions: $[FA]_0 = 500$ ppbv; blue LED ($\lambda = 460$ nm) intensity of 2.0 mW/cm²; RH 65%; reaction temperature of 30 °C.

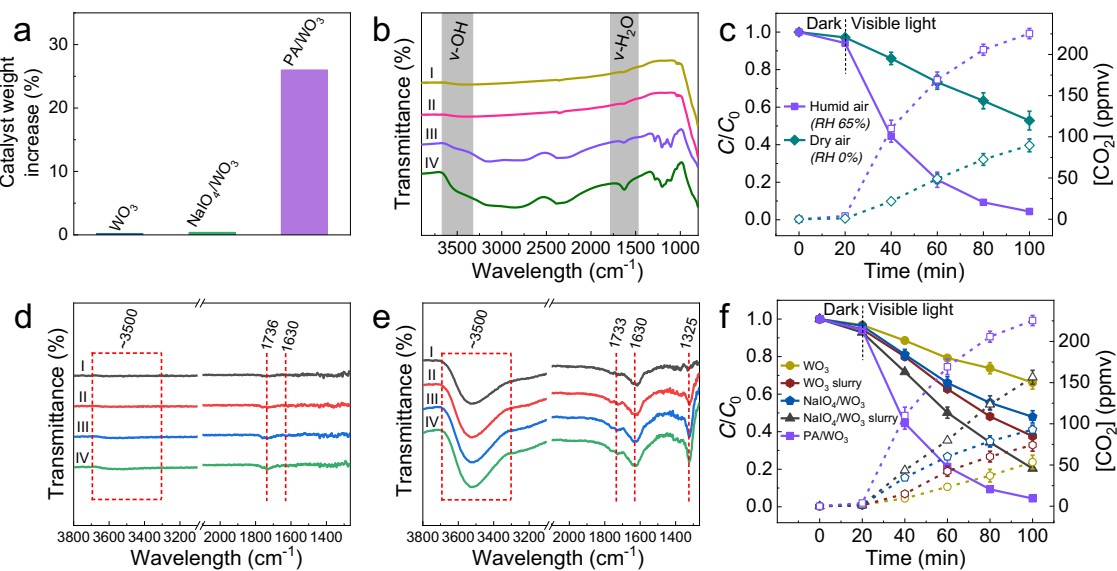

**Fig. 3 Formation and effects of the in situ formed water layer.** **a** Catalyst weight increase of PA/WO₃, NaIO₄/WO₃, and bare WO₃ samples after exposing each catalyst under humid air (RH 65%) for 30 min. RH is defined as relative humidity. **b** FT-IR spectra of bare WO₃ (I) before and (II) after exposure to humid air, and those of PA/WO₃ (III) before and (IV) after exposure to humid air for 30 min. **c** PCD activity of PA/WO₃ for acetaldehyde (AA) degradation after exposure to dry and humid air for 30 min. In situ DRIFT spectra for **d** PA/WO₃ after exposing the sample to 300 ppmv AA/dry air stream and **e** PA/WO₃ in 300 ppmv AA/RH 65% air stream for (I) 10 s, (II) 1 min, (III) 3 min, and (IV) 5 min. The spectra of the dry PA/WO₃ surface were collected and used as the background. **f** PCD activity of bare WO₃, NaIO₄/WO₃, and their slurries containing 26 wt% water for the degradation of AA. The dashed lines with open symbols represent the CO₂ concentration generated from AA degradation. Error bars are defined as standard deviation. Experimental conditions: $[AA]_0 = 120$ ppmv; visible light ($\lambda > 420$ nm) intensity of 2.2 mW/cm²; sample amount of 50 mg (with 13 mg of extra water in the case of WO₃ slurry and NaIO₄/WO₃ slurry); reaction temperature of 30 °C.

$3600-3200 \, cm^{-1}$ ($\nu$-OH), while the signal of bending vibration of adsorbed water appeared at $\sim 1630 \, cm^{-1}$ ($\nu$-$H_2O$)[41]. The signals of both ($\nu$-OH) and ($\nu$-$H_2O$) for bare $WO_3$ remained almost unchanged after exposure to humid air, which confirms that bare $WO_3$ surface does not induce significant water adsorption. However, these signals on PA/$WO_3$ were markedly enhanced after exposure to humid air, which indicates the condensation of water. The strong hydrogen bonding between the hydroxyl groups of PA and water molecules should make the condensation of water vapor highly exothermic at ambient conditions, where the negative $\Delta H$ outweighs the entropy decrease of water vapor condensation to make the overall condensation process thermodynamically spontaneous ($\Delta G < 0$). Therefore, the interfacial characteristics in the catalyst surface region of PA/$WO_3$ should be changed from the two-phase gas–solid interaction to the three-phase gas–liquid–solid interaction after the PA-induced condensation of the water layer.

To clarify the effect of water, the PCD activities of PA/$WO_3$ were compared between the dry (0% RH) and humid air (65% RH) conditions (Fig. 3c). Both the degradation of AA and the accompanying production of $CO_2$ over PA/$WO_3$ were significantly reduced under the dry air condition, which confirms that the role of water vapor is critical in controlling the overall PCD process. However, if not in the dry condition, the humidity variation ranging in RH 40–90% has a minor influence on the PCD of AA on PA/$WO_3$ (see Supplementary Fig. 9). It should be also noted that PA/$WO_3$ showed higher PCD activity than bare $WO_3$ even in dry air (Figs. 1a vs. 3c). The adsorption of VOCs is the first step in gas–solid photocatalytic reaction, and the adsorption properties of VOCs often significantly affect the PCD efficiency[42]. In the case of PA/$WO_3$ in humid air condition, the spontaneous formation of the liquid water layer on the catalyst surface should hinder the direct contact between VOC molecules and the $WO_3$ surface, which makes the reaction medium involve all three phases. As a result, the first step in PCD should be changed from the adsorption of AA on the $WO_3$ surface to the dissolution of AA in the surface water layer. Hydrophilic AA is highly soluble in water at room temperature[43]. The dark adsorptive removal of AA over PA/$WO_3$ was compared after 30 min equilibration under dry air and humid air (65% RH). It is found that 6.5% and 18.6% of AA is adsorbed on the catalyst surface within 100 min under dry and humid air conditions, respectively, which indicates that the existence of a water layer promoted the uptake of AA on the catalyst surface.

The relation between AA and the surface water layer was further investigated by in situ DRIFT spectroscopy under dark conditions. When exposing PA/$WO_3$ to 300 ppmv AA/dry airflow (see Fig. 3d), no peaks corresponding to $\nu$-OH ($3600-3200 \, cm^{-1}$) and $\nu$-$H_2O$ ($\sim 1630 \, cm^{-1}$) were found, which indicates that the formation of the surface water layer is negligible. The small peak located at $1736 \, cm^{-1}$ ($\nu$-$C=O$ vibration mode in aldehydes) is assigned to AA adsorbed via hydrogen bonding with a surface OH group[44]. On the other hand, the $\nu$-OH ($3600-3200 \, cm^{-1}$), $\nu$-$H_2O$ ($\sim 1630 \, cm^{-1}$), and $\nu$-$C=O$ ($1733 \, cm^{-1}$) peaks all increased significantly when exposing PA/$WO_3$ to 300 ppmv AA/65% RH airflow, which indicates the formation of the surface water layer and the enrichment of AA (see Fig. 3e). Moreover, a distinct peak corresponding to C–O stretching vibration ($\nu$-C–O) for carboxylic acids ($1325 \, cm^{-1}$)[45,46] gradually appeared with time. This implies that the in situ water layer formation facilitates not only adsorption/dissolution of AA but also partial pre-oxidation of AA, which subsequently accelerates the PCD process under irradiation.

To further confirm the role of the surface water layer, an aliquot of water was added to $WO_3$ powder to make a slurry (with and without $NaIO_4$): 13 mg water was well mixed with 50 mg $WO_3$ or

50 mg $NaIO_4$/$WO_3$ (w/w: 1/1) to form slurries. The resulting slurry photocatalysts and the corresponding dry photocatalysts were compared for the PCD of AA (see Fig. 3f). The slurry photocatalysts (with and without $NaIO_4$) showed higher PCD activity than the corresponding dry samples and the $NaIO_4$/$WO_3$ slurry exhibited higher PCD activity than pure $WO_3$ in either slurry or dry state. This confirms that the high activity of PA/$WO_3$ should be ascribed to the combined action of $IO_4^-$ and water. The PA acidity in PA/$WO_3$ may play a role as well since the pH of the PA solution was 1.5 whereas that of $NaIO_4$ solution was 4.6. To test the acidity effect, the pH of $NaIO_4$ solution was adjusted to 1.5 with adding iodic acid when preparing $NaIO_4$/$WO_3$ and the PCD activity of the acidified $NaIO_4$/$WO_3$ slurry was much higher than that of $NaIO_4$/$WO_3$ slurry (see Supplementary Fig. 10). Considering that iodic acid alone did not promote the PCD activity of $WO_3$, the above result implies that the PA acidity in the water layer should contribute to the high PCD activity of PA/$WO_3$. This might be related to the fact that the reduction of periodate is favored at an acidic condition ($IO_4^- + 2H^+ + 2e^- \rightarrow IO_3^- + H_2O$). Therefore, the water-rich surface over PA/$WO_3$ should scavenge photogenerated holes more efficiently to produce •OH and AA molecules that are dissolved and concentrated within the thin water layer should undergo the subsequent PCD reactions in the solid–water interface. This process should be much faster than that in the traditional gas–solid interfacial PCD reaction since the AA molecules in the surface water layer is far more concentrated and oxidized than those in the gas phase. The synergy between $IO_4^-$ and the water layer is further discussed in the following section.

The structural and surface properties of PA-treated $WO_3$ were evaluated by using XRD, FE-SEM, $N_2$ adsorption–desorption, XPS, DRS techniques, and density functional theory (DFT) calculations. As shown in Fig. 4a, the position of diffraction peaks in the XRD pattern of PA/$WO_3$ was consistent with those of monoclinic $WO_3$ (JCPDS-ICDD card #75-2072) and $HIO_4\cdot2H_2O$ (JCPDS-ICDD card #74-0334). No characteristic peaks of any other phases were observed, indicating that the original crystal form of $WO_3$ and PA was not affected in the prepared PA/$WO_3$ sample. In addition, PA treatment had a negligible effect on the BET surface area ($S_{BET}$), pore structure, and morphology of $WO_3$ (see Supplementary Figs. 11 and 12). To evaluate the light absorption capacity of samples, the UV–vis spectra of $WO_3$ and PA/$WO_3$ are compared in Fig. 4b. The light absorption capacity of $WO_3$ was clearly enhanced in the range of 300–450 nm after combining with PA. This PA-enhanced absorption is more clearly seen in the difference DRS spectra (see the inset of Fig. 4b). The extra light absorption should not be ascribed to the formation of an in situ water layer on PA/$WO_3$ since the absorption spectrum of $WO_3$ slurry (26 wt% water content) was a little different from that of dry $WO_3$ powder. Therefore, the enhanced absorption in the 300–450 nm region (inset of Fig. 4b) where PA itself is optically transparent, should be ascribed to the surface interaction between PA and $WO_3$. The XPS analysis was also employed to investigate the interaction between PA and $WO_3$. In the measured W 4f XPS spectra (Fig. 4c), the bare $WO_3$ sample shows two binding energy (BE) peaks at 35.5 and 37.7 eV, which are assigned to $4f_{7/2}$ and $4f_{5/2}$ of $W^{6+}$, respectively[47]. The two peaks of PA/$WO_3$ were shifted by $\sim 0.2$ eV to higher BE in comparison to those of bare $WO_3$. Several studies showed that the inorganic acid treatment of $TiO_2$ shifts the metal BE higher because of the strong interaction between the acid anions and metal cations[48–50]. It can be reasonably inferred that the oxygen atom in the PA molecule could be directly complexed with W cation on the surface of $WO_3$ in the form of "W–O–I–$(OH)_n$". In this complex, the electron density on the W cation should be further decreased by the strong electron-withdrawing character of the iodine cation ($I^{7+}$), leading to the slight increase in the BE of W 4f. The I 3d XPS spectra

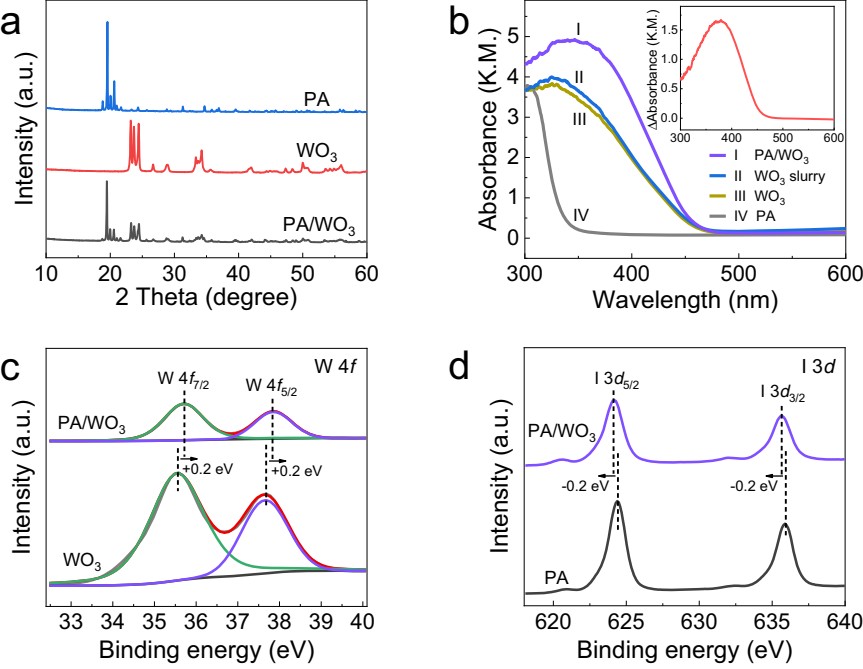

**Fig. 4 Characterizations of PA, WO₃, and PA/WO₃. a** X-ray diffraction spectra. **b** Diffuse reflectance UV–visible absorption spectra (DRS) (Inset is the difference DRS spectrum which is obtained from the spectrum I minus the spectrum III). XPS spectra of **c** W 4*f* and **d** I 3*d* band.

(shown in Fig. 4d) may further support such a conclusion. The BEs of I $3d_{5/2}$ and I $3d_{3/2}$ for PA are at 624.4 and 635.9 eV, respectively, but those of PA/WO₃ exhibit a negative shift by ~0.2 eV, which indicates the increased electron density on iodine. This suggests that the charge transfer in the "W–O–I–(OH)$_n$" complex is induced by the "electron-withdrawing effect" of the iodine cation to decrease the electron density on W but to increase that on I. Such charge transfer complex formation on the WO₃ surface not only facilitates the separation of photogenerated electron–hole pairs but also promotes the light absorption ability of WO₃[51–53], as supported by the enhanced light absorption of PA/WO₃ in Fig. 4b. Moreover, the DFT calculation that was carried out to investigate the interaction between PA and WO₃ surface shows that the calculated adsorption/binding energy of PA on WO₃ surface is −3.7 eV, a high value which usually implies the formation of strong chemical bonds[54]. The charge density difference analysis shows that there is an electron-depleted region on the WO₃ surface and an electron-gaining region around the iodine center. This clearly indicates that the charge is transferred from the WO₃ surface to PA, which is consistent with the XPS results (see Supplementary Fig. 13).

The role of dioxygen was also investigated. The PCDs of AA over PA/WO₃ and bare WO₃ were conducted under different O₂ concentrations (Fig. 5a and b). In general, the PCD rate is faster with producing more CO₂ at higher O₂ concentration (as shown in Fig. 5b) since dioxygen serves as the main electron acceptor in gas–solid interface PCD reaction with the concurrent formation of reactive oxygen species (e.g., O₂•⁻, HO₂•)[55]. This effectively transfers electrons to facilitate the charge carrier separation, and the resulting reactive oxygen species play a key role in oxidizing VOCs. However, the PCD of AA over PA/WO₃ was a little dependent on O₂ concentration and 120 ppmv of AA could be completely removed even in the absence of O₂ although the concurrent production of CO₂ was a little reduced without O₂ (Fig. 5a). This indicates that PA is a much stronger electron acceptor than dioxygen [$E^0$ (IO₄⁻/IO₃⁻) = +1.623 V$_{NHE}$, $E^0$ (O₂/O₂•⁻) = −0.33 V$_{NHE}$, $E^0$ (O₂/HO₂•) = −0.05 V$_{NHE}$][13,35,36]. The presence of O₂ did not enhance the degradation rate of AA

but moderately increased CO₂ generation. This is consistent with the fact that O₂ is needed as a reagent for the complete mineralization of aldehydes into CO₂[56].

The role of PA as the main electron acceptor implies that it should be consumed with irradiation time. Multi-cycle PCD experiments of AA over PA/WO₃ were conducted (Fig. 5c), which indeed exhibited a gradual decrease in the PCD rate. However, the PCD activity was maintained relatively high for a long irradiation period. In a comparison of the first and the fifth cycles, the removal of AA in 80 min decreased from 95.6% to 78.5%, and the fraction of mineralized AA (estimated from the CO₂ production) in 80 min was reduced from 93.9% to 70.3%. Even after the fifth cycle, the PCD activity was far higher than bare WO₃ (Fig. 1). FE-SEM, TEM, and XPS results showed that the morphology and surface chemical state of WO₃ in PA/WO₃ were little changed after the PCD reaction of AA (see Supplementary Figs. 12 and 14). The analysis of the iodine-containing species in PA/WO₃ during the cyclic experiments (Fig. 5d) found that the IO₄⁻ portion decreased from 100% in the fresh PA/WO₃ to 90.5% after the first cycle, and further to 59.3% after the fifth cycle of AA degradation. At the same time, the IO₃⁻ portion increased from 0% to 9.5% after the first cycle, and further to 40.7% after the fifth cycle. On the other hand, no I⁻ was detected throughout the multicycle experiments. It seems that the photocatalytic reaction of IO₄⁻ with CB electrons leads to the production of IO₃⁻ as a main product on the visible light-irradiated PA/WO₃. The possible reoxidation of IO₃⁻ to IO₄⁻ can be ruled out since the control PCD tests of AA using NaIO₃/WO₃ and HIO₃/WO₃ showed no production of IO₄⁻. Therefore, the gradual decline of catalyst activity of PA/WO₃ in Fig. 5c should be ascribed to the consumption of IO₄⁻. To regenerate the photocatalyst, the used PA/WO₃ was washed with water and reloaded with 0.11 mol/L PA solution by following the same procedure as in the preparation of the fresh PA/WO₃. The regenerated PA/WO₃ catalyst almost fully recovered its original activity (see Fig. 5c, VI), which confirms the critical role of PA as an electron acceptor in the PCD of AA.

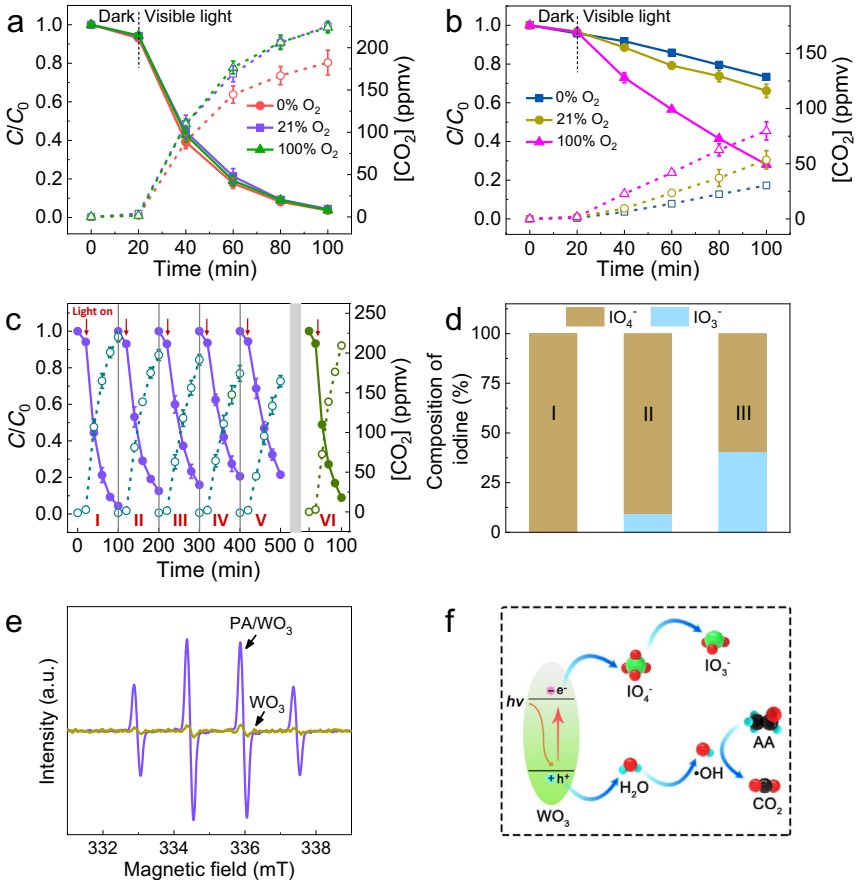

**Fig. 5 Effects of O₂ and PA.** Effect of $O_2$ concentration on the PCD of acetaldehyde (AA) over **a** PA/WO₃ and **b** bare WO₃. **c** Repeated PCD cycles of AA degradation and the concurrent production of $CO_2$ over PA/WO₃ (I, 1st cycle; II, 2nd cycle; III, 3rd cycle; IV, 4th cycle; V, 5th cycle), (VI) PA/WO₃ was regenerated after 5th cycle. Error bars are defined as standard deviation. **d** Relative distribution of iodine species in (I) fresh PA/WO₃, PA/WO₃ used after (II) 1st PCD cycle and (III) 5th PCD cycle of AA degradation in **c**: $IO_4^-$ (brown) and $IO_3^-$ (cyan). **e** EPR spectra probing the photogeneration of •OH adduct (•OH-DMPO) in an aqueous catalyst slurry containing DMPO ($C_0 = 10$ mM). **f** Proposed PCD mechanism of AA. The dashed lines with open symbols represent the $CO_2$ concentration generated from AA degradation. Experimental conditions: $[AA]_0 = 120$ ppmv; visible light ($\lambda > 420$ nm) intensity of 2.2 mW/cm²; sample amount of 50 mg; RH 65%; reaction temperature of 30 °C.

On the other hand, as for the hole transfer part, the photocatalytic production of OH radicals is one of the primary mechanisms by which photocatalysts degrade VOCs. The ability of PA/WO₃ to produce •OH was tested using a chemical trapping method using DMPO as a spin trap reagent[57]. After 10 min of blue LED irradiation (460 nm, 2.0 mW/cm²), EPR signals for the •OH-DMPO adduct appeared as shown in Fig. 5e. Note that PA/WO₃ produced a prominent signal whereas bare WO₃ generated an insignificant one. In general, bare WO₃ is not an efficient photocatalyst because of the lower CB edge (~0.4 $V_{NHE}$)[58] that does not provide a sufficient potential to reduce $O_2$ [$E^0$ ($O_2/O_2•^-$) = −0.33 $V_{NHE}$ and $E^0$ ($O_2/HO_2•$) = −0.05 $V_{NHE}$][13,20]. The inability of $O_2$ to scavenge CB electrons in WO₃ results in fast recombination and lower photocatalytic activity. After introducing PA, the periodate ions in the in situ formed surface water layer can scavenge CB electrons, which subsequently retards the charge recombination and makes more holes available for the production of •OH that should be the primary oxidant in the PCD process. The photocatalytic oxidation mechanism (see Fig. 5f) can be briefly proposed as follows (Eqs. (1)–(5)):

$$WO_3 + h\nu \rightarrow h_{vb}^+ + e_{cb}^- \tag{1}$$

$$HIO_4 (PA) \rightarrow H^+ + IO_4^- \text{ (in \textit{in situ} water layer)} \tag{2}$$

$$IO_4^- + 2H^+ + 2e_{cb}^- \rightarrow IO_3^- + H_2O \tag{3}$$

$$H_2O + h^+ \rightarrow •OH + H^+ \tag{4}$$

$$AA + •OH + O_2 \rightarrow\rightarrow CO_2 \tag{5}$$

**Selective PCDs for hydrophilic vs. hydrophobic VOCs.** Other VOCs including methanol (MeOH), isopropanol (IPA), acetone (AT), n-pentane (C5), dichloromethane (DCM), n-chloropropane (ClC₃), and toluene (Tol) were also tested for their PCDs by bare WO₃ and PA/WO₃. As shown in Fig. 6a and b, the PCD behaviors are clearly different depending on the kind of target VOCs. Compared with bare WO₃, PA/WO₃ increased the PCD rate constant ($k_d$) by 6.7, 6.5, and 7.5 times for the degradation of IPA, AT, and AA, respectively. Accordingly, the mineralization efficiency of IPA, AT, and AA over PA/WO₃ was 5.2, 5.2, 4.2 times higher than that over bare WO₃, respectively. Note that the PCD of MeOH was even more dramatically enhanced than that of AA on PA/WO₃: $k_d$ for MeOH on PA/WO₃ ($122.71 \times 10^{-3}$ min⁻¹) was 27.8 times higher than that of bare WO₃ ($k_d = 4.42 \times 10^{-3}$ min⁻¹). The PCDs of all the hydrophilic VOC molecules were markedly enhanced on PA/ WO₃, which implies that they can be easily dissolved and

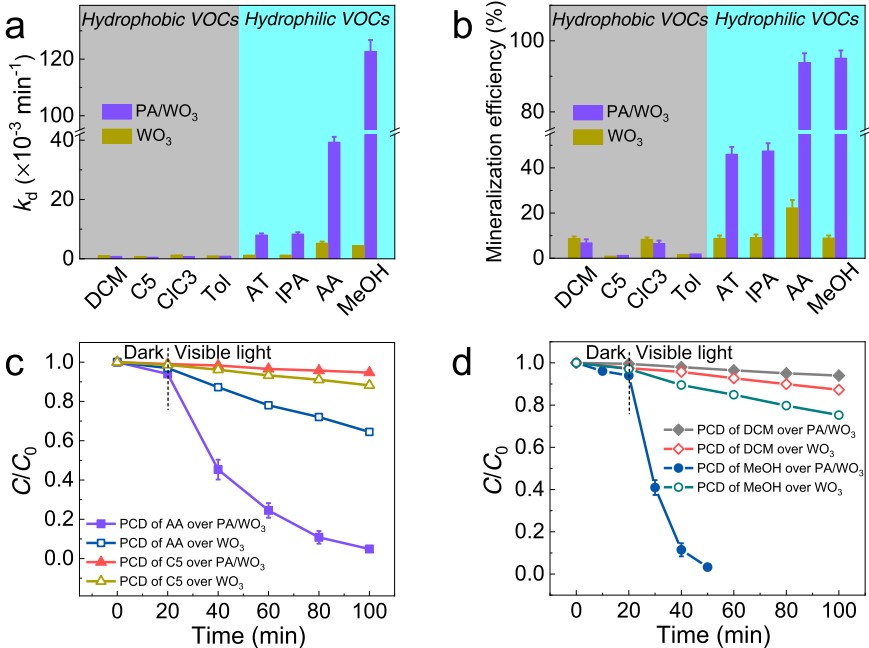

**Fig. 6 Selective removal of hydrophilic volatile organic compounds (VOCs) over PA/WO₃. a** PCD rate constants ($k_d$) and **b** the mineralization efficiencies of different VOCs over bare WO₃ and PA/WO₃. PCD of the mixture of **c** AA and C5 and **d** MeOH and DCM. Error bars are defined as standard deviation. Experimental conditions: [VOC]₀ = 120 ppmv; visible light ($\lambda > 420$ nm) intensity of 2.2 mW/cm²; sample amount of 50 mg; RH 65%; reaction temperature of 30 °C. AA acetaldehyde, IPA isopropanol, AT acetone, MeOH methanol, DCM dichloromethane, C5 n-pentane, ClC₃ n-chloropropane, Tol toluene.

concentrated in the in situ formed surface water layer to facilitate their PCD. However, the situation was markedly different for the PCDs of hydrophobic VOC molecules. The PCD rates of DCM, C5, Tol, and ClC₃ were very low over bare WO₃ ($k_d < 2.0 \times 10^{-3}$ min$^{-1}$) and the corresponding mineralization efficiencies of these VOCs were lower than 10%. Unlike the case of hydrophilic VOCs which exhibited the dramatic PA-enhanced effect, PA/WO₃ slightly hindered the PCDs of hydrophobic VOCs, compared with bare WO₃. The water layer on PA/WO₃ should not dissolve hydrophobic VOC molecules and their PCDs should be limited under such conditions. The markedly contrasting PCD behaviors of PA/WO₃ between hydrophobic and hydrophilic VOCs should be ascribed to their different solubility in the in situ formed surface water layer.

The selective PCDs in VOC mixtures were also investigated under visible-light irradiation ($\lambda > 420$ nm), as shown in Fig. 6c and d. With PA/WO₃, more than 95% of AA could be removed in the AA/C5 mixture after 80 min of PCD while the concentration of C5 was almost unchanged. In the same manner, only MeOH could be selectively removed in MeOH/DCM mixture. On the other hand, PCD over bare WO₃ removed 35.5% of AA and 11.8% of C5 in AA/C5 mixture in 80 min; 24.7% of MeOH and 12.8% of DCM in MeOH/DCM mixture. This clearly shows that PA/WO₃ has good selectivity for the degradation of hydrophilic VOCs while bare WO₃ photocatalyst has much lower selectivity. It is quite clear that the presence of an in situ formed water layer on PA/WO₃ greatly facilitates the selective removal of hydrophilic VOC molecules while keeping hydrophobic VOCs intact in the hydrophilic–hydrophobic VOC mixture. As a result, the highly enhanced and selective removal of hydrophilic VOCs is enabled by the PA-treatment of WO₃ photocatalyst which induces the in situ formation of the water layer on the catalyst surface. The role of in situ water layer formation on the PCDs of VOCs has not been previously recognized and needs to be further investigated for the selectivity control of VOCs degradation.

## Discussion

The photocatalytic oxidation processes have been extensively investigated for the purification of polluted air and water. A variety of photocatalyst modification methods have been proposed and tested to improve the performance of PCD while many methods need complicated preparation procedures or costly materials such as noble metals. In this work, a very simple and inexpensive method of photocatalyst modification is proposed and demonstrated for the highly enhanced and selective degradation of VOCs. PA coating on photocatalysts proved to be an effective method to greatly enhance the photocatalytic activity of WO₃, BiVO₄, and N-TiO₂ for the degradation of AA (a common indoor air pollutant) under visible light. In particular, PA/WO₃ exhibited excellent performance in PCD of various hydrophilic VOCs. Under visible-light irradiation ($\lambda > 420$ nm), the PCD rate constant ($k_d$) of IPA, AT, AA, and MeOH over PA/WO₃ was 6.7, 6.5, 7.5, and 27.8 times higher than that of bare WO₃. The hygroscopic property of PA enables the photocatalyst to form in situ surface water (self-wetting) layer in ambient air while the periodate ions in the in situ water layer serve as efficient electron acceptors on WO₃ under visible light. In addition, PA formed surface complexes on WO₃ surface not only to enhance the visible light absorption capacity but also to suppress the charge pair recombination by making PA scavenge CB electrons more efficiently. The multiple roles of PA induce a water layer on the photocatalyst surface, dissolve and concentrate hydrophilic VOCs in the water layer, and make more holes available to produce OH radicals. As a result, PA/WO₃ photocatalyst is even more active than Pt-loaded WO₃ (a popular but expensive visible-light active photocatalyst) for the degradation of AA.

PA is proposed as a low-cost component to replace costly Pt cocatalyst for the removal of hydrophilic VOCs. PA is nonvolatile and comparatively nontoxic and its release into the air can be safely neglected. AA degraded by PA/WO₃ photocatalyst generated no detectable volatile byproducts. To check the possible

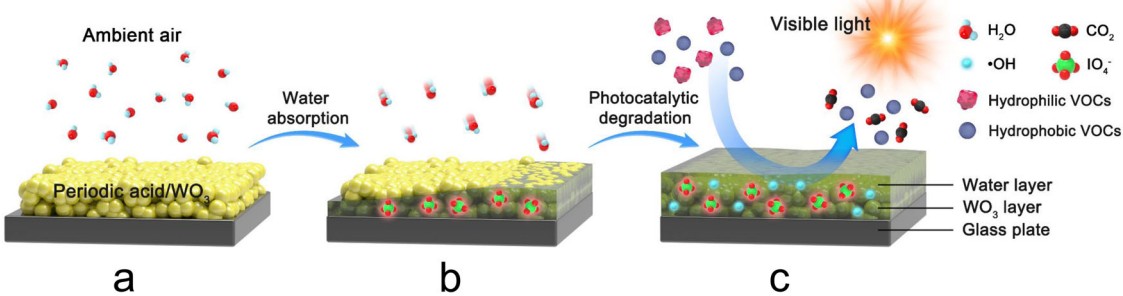

**Fig. 7 Schematic illustration of the working mechanism of PA/WO₃ photocatalysis for the selective PCD. a** $WO_3$ is coated with periodic acid (PA) to form PA/WO₃. **b** In situ surface water layer is formed due to the hygroscopic property of PA in ambient air while the periodate ions ($IO_4^-$) in the in situ water layer serve as efficient electron acceptors. **c** Under visible light, the photogenerated electrons are rapidly scavenged by $IO_4^-$ while holes produce •OH radicals which subsequently degrade hydrophilic VOCs dissolved into the in situ water layer. However, hydrophobic VOCs remain intact during photocatalysis due to the "water barrier effect".

generation of volatile iodine species (e.g., $I_2$, HOI) from the transformation of PA during the PCD of AA on PA/WO₃, the treated air was absorbed by phenol solution (1 mM) and analyzed for iodophenol (a product that should be generated from the reaction with volatile iodine species) by HPLC[59]. No iodophenol was detected, which indicated that no volatile iodine species was present in the treated air. No other gaseous organic products (e.g., FA, acetic acid) were found during the PCD of AA, which indicates the rapid mineralization of AA. The presence of a surface water layer can trap any hydrophilic intermediates and degrade them within the water layer without emitting them into the air phase. This self-wetting tri-phasic (air/water/catalyst) photocatalytic system facilitates the complete degradation of hydrophilic VOCs by providing the in situ water layer where the hydrophilic intermediates/byproducts are more efficiently retained and degraded in the aqueous phase whereas the common biphasic photocatalysis (air/catalyst) often generates gaseous intermediates.

From a practical point of view, a notable advantage is that the uptake of water vapor onto PA/WO₃ and the subsequent drying are reversible (as shown in the Supplementary video). This makes the wetting and drying process repeated depending on the humidity condition of the ambient air. As wetting a large surface is not convenient for practical applications, it is suggested that PA/WO₃ be employed as a replaceable filter component in air purifiers (especially for the removal of aldehydes such as AA and FA), not to be coated over the support with a large surface area. The reaction stoichiometry indicates that 5 moles of PA are needed to degrade 1 mole of AA ($CH_3CHO + 5IO_4^- \rightarrow 2 CO_2 + 5IO_3^-$). The cost of PA replacement should not be a problem because of its low price (99.9% purity, ~10.5\$/kg in the USA). Our analysis showed that 40.7% of initial PA (25 mg PA in 50 mg PA/WO₃) was reduced to iodate along with mineralizing 0.29 mg AA during five PCD cycles of AA (see Fig. 5d). This corresponds to the consumption of 1 mole PA for the degradation of 0.15 mole AA, which is close to the theoretical value (5:1). Based on this ratio, we estimate that 1 g PA is consumed to purify 317 m³ of indoor air contaminated with [AA] = 90 μg/m³ (10 times higher than the US EPA reference concentration for chronic inhalation exposure[60], 9 μg/m³). Moreover, the used filter can be washed with water and regenerated by replenishing PA or by recycling iodate back to periodate via electrochemical[61] or other economical chemical oxidation methods. Alternatively, a photoelectrochemical (PEC) filter device that can regenerate in situ periodate as soon as it is converted into iodate under irradiation in the PA/WO₃ filter plans to be developed in a further study. To further investigate the effect of the geometric surface area of the coated photocatalyst, the PCD activity

(for removing 500 ppbv FA in 1.5 L air) of PA/WO₃ was compared between the different geometric coating areas of 1 cm² vs. 4 cm² (see Supplementary Fig. 15). The PCD activity little decreased with decreasing the catalyst coating area from 4 to 1 cm², which shows that the ratio of treated air volume to catalyst coating area can reach over 1.5 L/cm². This clearly demonstrates that PA/WO₃ has a good capacity in purifying large volumes of air. In practical applications, this ratio is expected to be further enhanced by optimizing the catalyst dosage and coating thickness. Specific design parameters (e.g., airflow rate, catalyst dosage, thickness and area of catalyst coating, etc.) need to be carefully adjusted according to the actual application purpose. These engineering parameters remain to be investigated in future works. It is worth noting that the PA/WO₃ photocatalyst demonstrated good applicability for a wide range of VOC concentrations (500 ppbv–700 ppmv).

On the other hand, the inability of PA/WO₃ to remove hydrophobic VOCs (e.g., DCM, C5, ClC₃, and Tol) is a serious limitation for general purpose applications. However, its ability to degrade hydrophilic VOCs selectively against hydrophobic VOCs can be exploited in a specific application (see Fig. 7). In addition, the in situ formed water layer can protect the photocatalyst surface from fouling with inhibiting the deposition of hydrophobic components (e.g., indoor particulate matters like cooking particles, various VOCs containing halogen/phosphorus/silicon found in indoor environments)[62–64]. As a viable strategy for exploiting in situ water layer formation in ambient air photocatalysis, the PA/photocatalyst may be combined in a hybrid air treatment system where hydrophilic VOCs (that can be rapidly degraded by photocatalysis) are selectively degraded by the PA/photocatalyst and then hydrophobic VOCs (that are more recalcitrant against photocatalysis) are removed by other methods such as adsorption and thermal catalysis. Such an approach combines the advantages of various technologies to develop a more efficient and economical method of controlling VOCs.

## Methods

**Materials**. The chemicals used in this study are as follows: Tungsten oxide (WO₃, nanopowder, Sigma-Aldrich), titanium dioxide (TiO₂, P25, nanopowder, Evonik), bismuth vanadate (BiVO₄, nanopowder, Alfa Aesar), PA ($HIO_4·2H_2O$, ≥99.0%, Sigma-Aldrich), iodic acid (HIO₃, ≥99.5%, Sigma-Aldrich), sodium periodate (NaIO₄, ≥99.8%, Sigma-Aldrich), sodium iodate (NaIO₃, 99%, Sigma-Aldrich), sodium iodide (NaI, ≥99.0%, Sigma-Aldrich), chloroplatinic acid ($H_2PtCl_6·xH_2O$, ≥99.9%, Sigma-Aldrich), MeOH (CH₃OH, 99.9%, Samchun Chemicals), IPA ((CH₃)₂CHOH, 99.5%, Sigma-Aldrich), AT (CH₃COCH₃, 99.98%, Burdick Jackson), DCM (CH₂Cl₂, ≥99.8%, Sigma-Aldrich), C5 (CH₃(CH₂)₃CH₃, ≥99.0, Sigma-Aldrich), ClC₃ (CH₃CH₂CH₂Cl, 99%, Alfa Aesar), 5,5-dimethyl-1-pyrro-line-N-oxide (DMPO, ≥ 98.0, Sigma-Aldrich). Tol (300 ppmv, N₂ balance), AA (1000 ppmv, N₂ balance), FA (100 ppmv, N₂ balance), high-purity synthetic air (79% N₂/21% O₂) were purchased from Deokyang Company. All chemicals were of

reagent grade and used as received without further purification. Ultrapure deionized water (18 MΩ cm) prepared using a Millipore system was used.

**Preparation of photocatalyst materials**. Commercial $WO_3$ powder sample was treated with PA. Typically, 0.25 g $WO_3$ was dispersed in 10 mL 0.11 mol/L PA solution under sonication. The mixture was stirred at room temperature for 12 h, and the obtained suspension was completely dried in an oven at 80 °C. To compare with PA/$WO_3$ sample, control $WO_3$ samples that were treated using $NaIO_4$, $NaIO_3$, $HIO_3$, or $NaI$ as an alternative reagent were also prepared under the same experimental conditions.

Two other visible light photocatalysts of N-doped $TiO_2$ (N-$TiO_2$) and $BiVO_4$ were also treated with PA and compared with $WO_3$ for the effects of PA treatment. N-$TiO_2$ was prepared by high-temperature nitridation of $TiO_2$[65]: commercial P25 powder was treated in a tubular furnace at 500 °C under $NH_3$ gas flow (150 mL/min) for 5 h. Then, 0.25 g of N-$TiO_2$ and $BiVO_4$ were treated with 0.11 mol/L PA solution under the same condition as that of $WO_3$. In addition, Pt/$WO_3$ was prepared using a photodeposition method as another standard visible light photocatalyst[23,27]. Typically, an aqueous suspension of $WO_3$ (0.5 g/L) was irradiated with a 200 W mercury lamp for 30 min in the presence of chloroplatinic acid as a Pt precursor and MeOH (1 mol/L) as an electron donor. The amount of platinum loading was fixed at 1 wt% for $WO_3$. The resulting Pt/$WO_3$ powder was collected by filtration and washed with deionized water.

**PCD experiments**. The PCD of VOCs was conducted under visible light irradiation from a mercury lamp (custom-made), which was filtered through a 420-nm cutoff ($\lambda > 420$ nm) filter. The filtered light intensity on the photocatalyst was measured to be 2.2 mW/cm$^2$ by a power meter (Newport 1918-R). A closed-circulation glass reactor (300 mL) with a quartz window (a radius of 3 cm) was used (see Supplementary Fig. 1). A magnetic bar was placed at the bottom of the glass reactor to stir the air. The reactor was connected to a gas chromatograph (GC-Agilent 6890 Plus) equipped with a methane converter, a Porapak R column, an automatic sampling valve using an air actuator, and a flame ionization detector. The relative humidity (RH) was adjusted to ~65% by bubbling air through a stainless-steel bottle containing deionized water. A heating device was used to maintain the temperature of the photocatalytic reactor at ~30 °C. To test the air-tightness of the reactor, it was filled with high-purity synthetic air (79% $N_2$/21% $O_2$) and exposed to ambient air. A negligible increase in $CO_2$ (<4 ppmv within 60 min) was detected, indicating good airtightness of the reactor system.

Photocatalyst powder (50 mg) was dispersed in water in a quartz glass sink which possessed a 2 cm × 2 cm groove to hold the catalyst component. The catalyst slurry was completely dried and placed in the reactor for PCD tests. Before each PCD experiment, the glass reactor was purged with high-purity synthetic air (79% $N_2$/21% $O_2$) and irradiated under the mercury lamp or LED lamp to degrade any organic impurities remaining on the photocatalyst surface. The cleaning irradiation continued until the photogeneration of $CO_2$ was not detected.

Target VOCs tested in this study include FA, AA, IPA, AT, MeOH, DCM, C5, ClC$_3$, and Tol. AA or Tol was introduced into the photocatalytic reactor through diluting the standard gas (1000 ppmv AA, 300 ppmv Tol in $N_2$). For other VOCs (IPA, AT, MeOH, DCM, C5, or ClC$_3$), a calculated amount of each liquid sample was injected into the reactor and subsequently vaporized into the gas phase. The initial concentration of VOCs was adjusted to 120 ppmv. After 20 min equilibration for complete dispersion and pre-adsorption of VOCs on the photocatalyst surface, the mercury lamp was turned on to initiate the PCD process. The removal of each VOC and the accompanying $CO_2$ production was monitored in real-time by using a GC. All the control experiments were conducted under the same condition.

The PCD experiments of FA were carried out using a bigger reactor (1.5 L) instead of a 300 mL reactor to demonstrate the photocatalytic air treatment on a larger scale. A 460 nm-emitting LED (Luna Fiber Optic Korea, ICN14D-096) was employed as a light source. Photocatalyst powder (1, 10, 50, and 100 mg) was dispersed in water in a quartz glass groove (2 cm × 2 cm), which was dried and placed in the PCD reactor. Before each PCD test, the glass reactor was purged with high-purity synthetic air (79% $N_2$/21% $O_2$) and irradiated under an LED lamp to clean the photocatalyst surface. FA was introduced into the reactor through a mass flow controller regulating the standard gas (100 ppmv FA in $N_2$) flow. The initial concentration of FA was adjusted at 500 ppbv, which is much lower than that of other VOCs (120 ppmv). This represents a more realistic test condition where FA is present in an indoor air environment. A photoacoustic gas monitor (LumaSense, INNOVA 1412i) was used to measure the concentrations of FA.

**Analysis and characterizations**. X-ray diffraction (XRD) patterns of the photocatalyst samples were collected using an X-ray diffractometer (PANalytical X'Pert diffractometer) using Cu Kα irradiation. Nitrogen adsorption–desorption isotherms were recorded at 77 K by using a BELSORP-MINI II (BEL-Japan, Inc.). Before the measurement, the sample was degassed at 423 K overnight. The specific surface area was calculated via a multipoint BET analysis of the $N_2$ adsorption isotherm. FE-SEM images were taken by JSM-7800 F prime microscope at National Institute for Nanomaterials Technology (NINT, Pohang, Korea). X-ray photoelectron spectroscopy (XPS) was conducted using a Thermo Scientific K-Alpha XPS with Al Kα ($h\nu = 1486.6$ eV) as the excitation source. Fourier transform

infrared (FTIR) spectra were obtained using an attenuated total reflectance-FTIR (ATR-FTIR) spectrometer (Thermo Scientific Nicolet iS50 FT–IR/ATR). Diffuse reflectance UV–visible absorption spectroscopy (DRS) was conducted using a spectrophotometer (Shimadzu UV-2401PC).

The quantitative analysis of iodine species, mainly iodate ($IO_3^-$) and periodate ($IO_4^-$) was carried out using HPLC (Agilent 1100) equipped with an Agilent Zorbax 300SB C-18 column and a diode-array detector. The typical eluent consisted of a binary mixture of 0.1% (v/v) phosphoric acid aqueous solution and acetonitrile (typically 70:30 v/v). The photocatalytic production of OH radicals was confirmed using an electron paramagnetic resonance (EPR) spin trapping method (ELEXSYS E580, Bruker Co.): 10 mM DMPO as a spin-trapping agent was added to the aqueous catalyst slurry under irradiation of blue LED (Luna Fiber Optic Korea, CWL 460 nm, 2.0 mW/cm$^2$). Real-time monitoring of the dynamic movement of catalyst particles and in situ formed water layer in the PA/$WO_3$ system was performed using a Leica DM 5000 B microscope equipped with a Leica DFC420 camera (Leica, Wetzlar, Germany).

In situ DRIFTS was performed using an FT-IR spectrometer (PerkinElmer, USA) equipped with a diffuse-reflectance cell (PIKE) with a ZnSe window. The catalyst sample was placed in the cell and pretreated at 100 °C to eliminate the effects of adsorbed water. AA of 300 ppmv was introduced into the cell by diluting the standard gas (1000 ppmv AA in $N_2$) with air. The RH was adjusted by bubbling the air through a stainless-steel bottle containing deionized water. In situ DRIFTS spectra were collected after exposing the samples to the flowing stream for 10 s, 1, 3, and 5 min, respectively.

**Computational details**. Density function theory (DFT) calculations were performed by using the CP2K package[66]. PBE functional[67] with Grimme D3 correction[68] was used to describe the system. Unrestricted Kohn–Sham DFT has been used as the electronic structure method in the framework of the Gaussian and plane waves method[69]. The Goedecker–Teter–Hutter (GTH) pseudopotentials[70], DZVPMOLOPT-GTH basis sets[69] were utilized to describe the molecules. A plane-wave energy cut-off of 500 Ry has been employed.

The charge density difference is defined as Eq. (6):

$$\Delta\rho = \rho_{mol}/_{sur} - \rho_{mol} - \rho_{sur} \tag{6}$$

where $\rho_{mol}/_{sur}$, $\rho_{mol}$, and $\rho_{sur}$ are the electron density of the molecule adsorbed on the surface, the molecule, and the surface, respectively. The BE is defined as Eq. (7):

$$E_b = E_{mol}/_{sur} - E_{mol} - E_{sur} \tag{7}$$

where $E_{mol}/_{sur}$, $E_{mol}$ and $E_{sur}$ are the energy of the molecule adsorbed on the surface, the molecule, and the surface, respectively.

**Calculation of apparent quantum efficiency (AQE)**. For the measurement of AQE, two monochromatic LEDs (Luna Fiber Optic Korea, 2.0 mW/cm$^2$) which emit light at ca. 365 and 460 nm were used in the PCD of AA. Except for the light source, all the conditions were the same as that in the PCD tests using the mercury lamp. The AQE of AA degradation was indirectly calculated based on the $CO_2$ generation rate within 40 min PCD reaction to rule out the effect of AA adsorption on the catalyst surface and reactor surface. The PCD of AA can be divided into the following two half-reactions (Eqs. (8)–(10)).

$$\text{Oxidation: } CH_3CHO + 3H_2O \rightarrow 2CO_2 + 10H^+ + 10e^- \tag{8}$$

$$\text{Reduction: } 2.5O_2 + 10H^+ + 10e^- \rightarrow 5H_2O \tag{9}$$

$$\text{Overall: } CH_3CHO + 2.5O_2 \rightarrow 2CO_2 + 2H_2O \tag{10}$$

The full degradation of an AA molecule leads to the production of two $CO_2$ molecules, which is a 10-electron oxidation. Therefore, the production of one $CO_2$ molecule needs a 5-electron oxidation, which should consume 5 photons: the AQE estimated from the $CO_2$ production measurement is based on this stoichiometry. The number of incident photons indicates the total number of photons reaching the surface of the catalyst during the reaction time, which were calculated by Eq. (11). The number of reacted photons indicates the number of photons that are utilized in transforming AA molecules into $CO_2$, which was calculated by Eq. (12). The AQE was calculated by Eq. (12)/Eq. (11). Such estimation of AQE was based on the assumption that AA molecules are fully mineralized to $CO_2$ with the negligible generation of intermediates. As the PCD of AA on PA/$WO_3$ accompanied the stoichiometric production of $CO_2$ (see Figs. 1, 3, 5), the formation of intermediates (if any) should be negligible in terms of the carbon mass balance.

$$\text{Number of incident photons} = \frac{E\lambda At}{hc} \tag{11}$$

$$\text{Number of reacted photons} = 5R_{CO_2}N_A t \tag{12}$$

$$\text{AQE} = \frac{\text{Number of reacted photons}}{\text{Number of incident photons}} \times 100\% \tag{13}$$

where $E$ is the intensity of LED (mW/cm$^2$); $\lambda$ is the wavelength of LED; $A$ is the irradiation area of photocatalysts; $t$ is the illumination/reaction time (40 min); $h$ is

the Planck constant, $6.626 \times 10^{-34}$ Js; $c$ is the velocity of light; $R_{CO2}$ is the $CO_2$ generation rate (mol/min); and $N_A$ is the Avogadro constant, $6.02 \times 10^{23}$/mol.

## Data availability

Source data are provided with this paper. All data are also available from the corresponding author on request.

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

## Acknowledgements
This work was supported by the Leading Researcher Program (NRF-2020R1A3B2079953) and Korea Research Fellowship Program (Grant No. 2018H1D3A1A02038503), which were funded by the Korea government (MSIT) through the National Research Foundation of Korea (NRF).

## Author contributions
F.H. conceived and performed the experimental studies, data-analysis, and wrote the manuscript. S.W. and M.W.C. conducted the in situ DRIFT test. W.J. helped with the experimental studies and equipment set-up. W.C. supervised the whole project, revised and reviewed the manuscript.

## Competing interests
The authors declare no competing interests.
