## [Peer Review File · Nature Communications]

Title: Self-wetting triphase photocatalysis for effective and selective removal of hydrophilic volatile organic compounds in airREVIEWER COMMENTS

Reviewer #1 (Remarks to the Author):

Comments:

In this research, the authors introduced in-situ water (self-wetting) layer on WO₃ by coating hygroscopic periodic acid (PA) to dramatically enhance the air-purifying photocatalytic activity. The presence of a thin surface water layer may solubilize and concentrate hydrophilic VOCs in it, which subsequently facilitates their degradation. Although the work has been carried out with care and the results reported in this paper are interesting and well organized, it is not sufficiently novel or significant. In summary, the novelty of the work, in terms of photocatalyst material and fundamental mechanism, is insufficient to be published in Nature Comm. Here are some comments.

1. The WO₃ is not a recognized favorable photocatalytic material (not highly efficient yet stable), also not sufficiently novel. Why did the author choose PA/WO₃ as photocatalyst?
2. As shown in the Supplementary Figure 2 and Supplementary Figure 3, the photocatalytic activities for PA/P25 under light irradiation is obviously higher than that of the PA/WO₃. Why did the author choose PA/WO₃ rather than PA/P25? In the direct ambient sunlight, P25 will also have more advantages.
3. Firstly, the author reported that the PA/WO₃ is almost ineffective for hydrophobic VOCs removal. Secondly, air humidity is an important physical factor that influences the photocatalytic air purification. As shown in Figure 2C, the influence of humidity on the photocatalytic performance is significant. Thus, the PA/WO₃ may be ill-suited for the practical air purification that contains various pollutants with varying humidity.
4. The concept that introduced a water layer on PA/WO₃ to dramatically enhance the air-purifying photocatalytic is not new. The fundamental mechanism is the same as the photocatalytic wet scrubbing process for air purification that has been widely reported (Ind. Eng. Chem. Res. 2010, 49, 3617–3622).
5. The authors indicate that PA act as an electron acceptor that can be that should be consumed with irradiation. The unsatisfactory stability was shown in the multi-cycle PCD experiments. Thus, the PA/WO₃ may be ill-suited for the practical air purification.
6. In situ DRIFTS tests should be conducted.

Reviewer #2 (Remarks to the Author):

Review Comments for NCOMMS-20-50917

" Self-wetting triphase photocatalysis with superb performance of air purification"

This manuscript presents an interesting study on the role of a periodic acid (PA) layer on photocatalytic materials such as WO₃ to enhance the photocatalytic degradation performance for hydrophilic volatile organic contaminants. PA provides hygroscopic properties resulting in air-water-catalyst triphasic system. The PA layer also serves as an effective electron acceptor that results in higher apparent photonic efficiencies. The authors determined apparent photonic efficiencies for degradation of acetaldehyde and some other hydrophilic organics that are obviously larger than those in the absence of

PA layer. I believe the authors present an interesting concept and convincing data. There are some aspects that need to be considered and addressed for a practical application of such process. Some comments are below.

- 1) Related to lines 82-84. What is the practicality of this for indoor air environment? Will this PA stay there for a long time to perform such function? Have the authors simulated this in a long-term operation performance test? In the field of indoor air quality, what could foul this layer? Will the presence of siloxanes be an issue?
- 2) Related to lines L131-136. While the authors somewhat discuss this, I feel more discussion is needed on the role of surface acidity. What is the role of acidity in this layer? How PA changes the surface charge of the respective photocatalysts? Do authors have quantitative analysis on this aspect?
- 3) Related to line L147-148. Relevant to point 1 above, how this will be relevant if you use this PA layer in a real indoor air application? Will you need to coat all indoor air surfaces?
- 4) Related to line L161. Do authors have more detailed insights through thermodynamic analysis with respect to condensation of water?
- 5) Related to lines L221-223. On the formation of surface charge transfer complex, can authors provide theoretical DFT calculations to examine further this effect?
- 6) Related to lines L238-247; 251-252, Fig. 4C and other relevant sections. Can authors describe how they plan to address the issue of reduced efficiency with time. Such coating should be effective for long term performance. If activity is reduced over time, how will the authors use them in a practical long-term indoor air application? So, how can authors solve the problem of reactivity of PA?
- 7) Supplementary Table 3. Can authors compare the AQE of their WO₃ with those of other such catalysts? Can authors make sure their efficiencies are not overestimated? For example these values reported in Fig. 4 appear higher than values reported in this Table for similar concentration of AA

Reviewer #3 (Remarks to the Author):

In this manuscript, the authors developed a novel strategy to introduce a periodic acid (PA) layer on WO₃ to dramatically enhance the photocatalytic activity for air purifying. They found that water vapor in ambient air could be condensed on the surface of WO₃ to form air/water/WO₃ structure. By this method, a high apparent quantum efficiency of 64.3% at 460 nm for acetaldehyde degradation was obtained. Further study suggested that periodic acid plays multiple roles as a water-layer inducer, a surface complexing ligand that enhances visible light absorption, and an electron acceptor. Moreover, other hydrophilic VOCs (methanol, isopropanol, and acetone) were also rapidly degraded by PA/WO₃, but hydrophobic VOCs were not. This study reports some new and interesting results and the presentation is also clear. However, some concerns should be addressed before this manuscript can be considered for publication.

(1) PA is used as an electron acceptor due to its strong oxidation, which can be consumed during the photocatalytic reaction (see Figure 4). Strictly speaking, PA/WO₃ is not a photocatalyst. Except for PA, is it possible that WO₃ also changes after the reaction? More characterization (TEM, SEM and XPS) of the composite samples before and after the reaction should be carried out. Moreover, PA is toxic, why did

the authors use it for air purifying?

(2) Isotope experiments should be carried out to confirm CO₂ products from VOC degradation, not from some contaminants.

(3) More calculation details on apparent quantum efficiency should be offered. For example, what does the number of reacted photons mean? how much are the t₁ and t₂, respectively. All of the oxidation products should be measured to calculate apparent quantum efficiency. Different from water splitting, VOC degradation reaction is a chain reaction, apparent quantum efficiency is possibly much higher than 100%. The authors should give some comments on so-called high apparent quantum efficiency in this study.

(4) From Figure 2 b, it is very difficult to obtain the concentration of H₂O or OH from the peak intensity of FT-IR. Other characterization should be used.

(5) In Figure 3b, is it possible that the reflectance spectra difference between WO₃ and PA/WO₃ comes from different aggregation states of WO₃ in different samples since the absorption edge does not change.

Response to Reviewer #1:

In this research, the authors introduced in-situ water (self-wetting) layer on WO_3 by coating hygroscopic periodic acid (PA) to dramatically enhance the air-purifying photocatalytic activity. The presence of a thin surface water layer may solubilize and concentrate hydrophilic VOCs in it, which subsequently facilitates their degradation. Although the work has been carried out with care and the results reported in this paper are interesting and well organized, it is not sufficiently novel or significant. In summary, the novelty of the work, in terms of photocatalyst material and fundamental mechanism, is insufficient to be published in Nature Comm. Here are some comments.

Specific Comments:

Comment 1. The WO_3 is not a recognized favorable photocatalytic material (not highly efficient yet stable), also not sufficiently novel. Why did the author choose PA/ WO_3 as photocatalyst?

Response: The reasons why we choose WO_3 as the photocatalyst are as follows:

(1) WO_3 is a representative visible light catalyst with the absorption range up to 480 nm, which can capture approximately 12% of the solar spectrum. In our previous work, it was found that WO_3 exhibits the highest visible light photocatalytic activity for the degradation of acetaldehyde than other common visible light photocatalysts, including N-TiO₂, CdS, TiO₂-WO₃, C-TiO₂, Ta₃N₅, and TaON (*see the figure below*). In this work, we also found that WO_3 photocatalyst is more active than BiVO₄ and N-TiO₂ (*see Figure 1A*). g-C₃N₄ was also tested only to exhibit a low activity.

Figure. Photocatalytic degradation of acetaldehyde (CH_3CHO) and the concurrent production of carbon dioxide (CO_2) on various visible active photocatalysts (obtained after 1 h reaction). (*Figure S8 in ACS Catal. 2016, 6, 8350–8360*)

(2) WO_3 is stable in oxidative and acidic condition (*Environ. Sci.: Nano, 2017, 4, 539-557; J. Environ. Chem. Eng., 2021, 9, 105018*), which is an important advantage because PA is highly acidic and oxidative. The stability of WO_3 in PA solution was confirmed using XRD, N₂ absorption-desorption experiments, FE-SEM and TEM (*see Figure 3A, Supplementary Figure 10, Supplementary Figure 11*).

(3) The main novelty of this work is focused on a self-wetting triphase photocatalytic system, so commercial catalysts were used to eliminate the interferences caused by preparation method,

morphology and structure differences. The PA effect was not limited to WO₃ but also observed for other typical photocatalysts such as TiO₂, WO₃, BiVO₄, N-TiO₂.

PA is suitable for air purifying purpose for the following reasons:

- (1) PA is comparatively non-toxic and nonvolatile (*Polym. Degrad. Stabil.*, 2012, 97, 816).
- (2) PA is cheap for industrial scale application (99.9% purity, 10.5 \$/Kg in USA).
- (3) PA has excellent electron accepting and oxidizing capacity.

Briefly, the following sentences were modified and added to clarify the reason of selection.

(p. 3) “WO₃ is one of the most frequently investigated photocatalysts with **notable** visible light activity ($E_g \approx 2.8$ eV)^{10,17,18}, **which is also stable in oxidative and acidic condition**¹⁹.”

(p. 16) “**PA is proposed as a low-cost component to replace costly Pt cocatalyst for the removal of hydrophilic VOCs. PA is nonvolatile and comparatively nontoxic and its release into the air can be safely neglected. The uptake of water vapor onto PA/WO₃ and the subsequent drying are reversible.**”

Comment 2. As shown in the Supplementary Figure 2 and Supplementary Figure 3, the photocatalytic activities for PA/P25 under light irradiation is obviously higher than that of the PA/WO₃. Why did the author choose PA/WO₃ rather than PA/P25? In the direct ambient sunlight, P25 will also have more advantages.

Response: Please note that PA/P25 was tested under UV light whereas PA/WO₃ was tested under visible light. The main merit of WO₃ is its visible light activity. In particular, WO₃ can be applied in the indoor environment where UV light intensity is negligible. The different light irradiation condition is further clarified in the captions of Supplementary Figure 2 as follows: “**Supplementary Figure 2.** PCD activities for (A) PA/P25 and P25 under irradiation of **UV mercury lamp (13.3 mW/cm²)** and for (B) **PA/P25 under irradiation of LED ($\lambda = 460$ nm or 365 nm, 2.0 mW/cm²).**”

To compare the photocatalytic activity under the same irradiation condition, the photocatalytic activities of PA/P25 and PA/WO₃ were additionally tested under LED irradiation of 365 nm and 460 nm as shown in Supplementary Figures 2 and 3. The following sentences were added to discuss this issue:

(p. 5) “**Interestingly, the photocatalytic activities of PA/P25 and PA/WO₃ are similar under LED irradiation of 365 nm while PA/WO₃ is far more active than PA/P25 under 460 nm LED irradiation (see Supplementary Figures 2 and 3).**”

Comment 3. Firstly, the author reported that the PA/WO₃ is almost ineffective for hydrophobic VOCs removal. Secondly, air humidity is an important physical factor that influences the photocatalytic air purification. As shown in Figure 2C, the influence of humidity on the photocatalytic performance is significant. Thus, the PA/WO₃ may be ill-suited for the practical air purification that contains various pollutants with varying humidity.

Response: We understand that the proposed PA/WO₃ photocatalytic system does have limitations for general purpose air purification. The main scientific novelty of this study is to find that the *in-situ*

formed water layer on PA/WO₃ selectively degrades hydrophilic VOCs with an unprecedented quantum efficiency (up to 64%!). Such phenomenon has not been reported to our knowledge. In addition, the unique PA/WO₃ system can have practical merits for its ability to remove hydrophilic VOCs only while retaining hydrophobic molecules. For example, the degradation of aromatic VOCs (mostly hydrophobic) is known to cause a rapid deactivation of photocatalysts (*Environ. Sci. Technol.* 2016, 50, 2556-2563; *Journal of Catalysis*, 2000, 196, 253-261; *Building and Environment*, 2012, 56, 329-334). Therefore, a strategy employing more targeted treatment for hydrophilic and hydrophobic VOCs can be proposed. We may develop a hybrid process which consists of the pre-treatment of the selective photocatalytic degradation of hydrophilic VOCs (*e.g.*, aldehydes, alcohols and ketones) using PA/WO₃ and the subsequent post-treatment method that mainly removes hydrophobic VOCs to make the overall treatment more efficient.

To represent the content of the work more properly, the title and the abstract were changed as follows.

Title: Self-wetting triphase photocatalysis for highly selective removal of hydrophilic volatile organic compounds in air

Abstract: “Here we devise a novel strategy to introduce an *in-situ* water (self-wetting) layer on WO₃ by coating hygroscopic periodic acid (PA) to dramatically enhance the photocatalytic removal of hydrophilic volatile organic compounds in air.”

As for the humidity effect, we additionally tested the RH effect (*see* Supplementary Figure 8) to show that the variation in the humidity level is not that significant. The following sentences were added to address this point:

(p. 8) “However, if not in the dry condition, the humidity variation ranging in RH 40-90% has minor influence on the PCD of AA on PA/WO₃ (*see* Supplementary Figure 8). It should be also noted that PA/WO₃ showed higher PCD activity than bare WO₃ even in dry air (Figure 1A vs. 2C).”

Supplementary Figure 8. PCD activity of PA/WO₃ for acetaldehyde (AA) degradation in humid air with different RH. The dashed lines with open symbols represent CO₂ generated from AA degradation. Experimental conditions: [AA]₀ = 120 ppmv; visible light ($\lambda > 420$ nm) intensity of 2.2 mW/cm²; sample amount of 50 mg; reaction temperature of 30 °C.

Comment 4. The concept that introduced a water layer on PA/WO₃ to dramatically enhance the air-purifying photocatalytic is not new. The fundamental mechanism is the same as the photocatalytic wet scrubbing process for air purification that has been widely reported (*Ind. Eng. Chem. Res.* 2010, 49, 3617–3622).

Response: We do not agree to this statement which is an exaggeration. The proposed PA/WO₃ system that utilizes the *in-situ* formed thin water layer under ambient air is completely different from the photocatalytic wet scrubbing process which employs a bulk aqueous suspension of photocatalyst particles. Although the wet scrubbing process also improves the photocatalytic removal of VOCs *via* constructing gas-liquid interface, our work has the notable innovations in the following points:

(1) A self-wetting photocatalytic system has been reported for the first time, and the water layer on the PA/WO₃ was *in situ* formed by absorbing water molecules in air whereas a dilute suspension of catalyst (mostly bare TiO₂) is employed in the wet scrubbing process (*Ind. Eng. Chem. Res.* 2010, 49, 3617-3622; *Sep. Purif. Technol.* 2012, 90, 196-203).

(2) PA plays multiple roles of a water-layer inducer, a surface complexing ligand that enhances visible light absorption, and a strong electron acceptor, which is used in the air purification system for the first time.

(3) In the wet scrubbing process, extra energy driving the stirrers and pumps is usually needed while PA/WO₃ is a much simpler self-working system, which can even work in the absence of oxygen (*see* Figure 4A);

(4) The mentioned photocatalytic wet scrubbing process operates under UV light irradiation, while PA/WO₃ reached the highest apparent quantum efficiency of 64.3% for the PCD of AA (the highest value ever reported) under visible light (460 nm) irradiation.

Comment 5. The authors indicate that PA act as an electron acceptor that can be that should be consumed with irradiation. The unsatisfactory stability was shown in the multi-cycle PCD experiments. Thus, the PA/WO₃ may be ill-suited for the practical air purification.

Response: Objectively speaking, the stability problem is a common and general problem in the photocatalytic air purification system, not unique to this system. In this study, the gradual decline of activity of PA/WO₃ in Figure 4C is due to the conversion of IO₄⁻ to IO₃⁻ in the multi-cycle PCD experiments (Figure 4D). However, the used PA/WO₃ can be easily regenerated by replenishing IO₄⁻ (*see* Figure 4C, f). The stability of WO₃ itself has little problem. To confirm the stability of WO₃, the Supplementary Figure 11 and Supplementary Figure 13 were added and the following sentence was added:

(p. 12) “FE-SEM, TEM and XPS results showed that the morphology and surface chemical state of WO₃ in PA/WO₃ were little changed after the PCD reaction of AA (*see* Supplementary Figure 11, Supplementary Figure 13).”

(Supplementary Figure 11) “The possible morphology changes of WO₃ after the PCD of AA was also investigated using FE-SEM and TEM. To eliminate the interference of abundant PA to the results, the PA/WO₃ used after PCD of AA was washed thoroughly with water (*denoted as* WO₃-used). As shown

above, no obvious change was found in the morphology of WO_3 -used compared with that of fresh WO_3 (namely WO_3 -untreated), indicating that the size and morphology of WO_3 were little changed during the PCD reaction of AA.”

Supplementary Figure 11. (A) FE-SEM and (B) TEM image of (a) WO_3 -untreated, (b) WO_3 -treated and (c) WO_3 -used.

Supplementary Figure 13. W 4f XPS spectra of the fresh and used PA/WO_3 samples.

Furthermore, the following sentences were added to discuss the PA issue:

(p. 16) “PA is proposed as a low-cost component to replace costly Pt cocatalyst for the removal of hydrophilic VOCs. PA is nonvolatile and comparatively nontoxic and its release into the air can be safely neglected. The uptake of water vapor onto PA/WO_3 and the subsequent drying are reversible. The gradual depletion of periodate is the critical problem that should be solved for practical applications. A possible solution is to employ PA/WO_3 as a replaceable filter component in air purifiers. The used filter

can be regenerated by recycling iodate back to periodate *via* electrochemical⁶⁰ or other economical chemical oxidation methods. Alternatively, a photoelectrochemical (PEC) filter device that can regenerate *in-situ* periodate as soon as it is converted into iodate under irradiation in the PA/WO₃ filter plans to be developed in a further study. Therefore, for practical applications, PA/WO₃ should be more suitable to be employed as a replaceable filter component, rather than to be coated over the support with a large surface area.”

Comment 6. In situ DRIFTS tests should be conducted.

Response: The in-situ DRIFT tests were additionally carried out (*see* Figure 2) and the following sentences were added:

(p. 8-9) “The relation between AA and the surface water layer was further investigated by *in situ* DRIFT spectroscopy under dark conditions. When exposing PA/WO₃ to 300 ppmv AA/dry air flow (*see* Figure 2D), no peaks corresponding to ν -OH (3600–3200 cm⁻¹) and ν -H₂O (~1630 cm⁻¹) were found, which indicates that the formation of surface water layer is negligible. The small peak located at 1736 cm⁻¹ (ν -C=O vibration mode in aldehydes) is assigned to AA adsorbed *via* hydrogen bonding with a surface OH group⁴³. On the other hand, the ν -OH (3600–3200 cm⁻¹), ν -H₂O (~1630 cm⁻¹), and ν -C=O (1733 cm⁻¹) peaks all increased significantly when exposing PA/WO₃ to 300 ppmv AA/65% RH air flow, which indicates the formation of surface water layer and the enrichment of AA (*see* Figure 2E). Moreover, a distinct new peak corresponding to C-O stretching vibration (ν -C-O) for carboxylic acids (1325 cm⁻¹)^{44,45} gradually appeared with time. This implies that the *in-situ* water layer formation facilitates not only adsorption/dissolution of AA but also partial pre-oxidation of AA, which subsequently accelerates the PCD process under irradiation.”

(p. 20) “*In situ* DRIFTS was performed using a FT-IR spectrometer (PerkinElmer, USA) equipped with a diffuse-reflectance cell (PIKE) with a ZnSe window. The catalyst sample was placed in the cell and pretreated at 100 °C to eliminate the effects of adsorbed water. AA of 300 ppmv was introduced into the cell through diluting the standard gas (1000 ppmv AA in N₂) with air. The relative humidity (RH) was adjusted by bubbling the air through a stainless-steel bottle containing deionized water. *In situ* DRIFTS spectra were collected after exposing the samples to the flowing stream for 10 s, 1 min, 3 min, and 5 min, respectively.”

Figure 2. (A) Catalyst weight increase of PA/WO_3 , $\text{NaIO}_4/\text{WO}_3$, and bare WO_3 samples after exposing each catalyst under humid air (RH 65%) for 30 min; (B) FT-IR spectra of bare WO_3 (a) before and (b) after exposing to humid air, and those of PA/WO_3 (c) before and (d) after exposing to humid air for 30 min; (C) PCD activity of PA/WO_3 for acetaldehyde (AA) degradation after exposing to dry and humid air for 30 min; *In situ* DRIFT spectra for (D) PA/WO_3 after exposing the sample to 300 ppmv AA/dry air stream and (E) PA/WO_3 in 300 ppmv AA/RH 65% air stream for (a) 10 s, (b) 1 min, (c) 3 min, and (d) 5 min. The spectra of the dry PA/WO_3 surface were collected and used as the background; (F) PCD activity of bare WO_3 , $\text{NaIO}_4/\text{WO}_3$, and their slurries containing 26 wt% water for the degradation of AA. The dashed lines with open symbols represent CO_2 concentration generated from AA degradation. Experimental conditions: $[\text{AA}]_0 = 120$ ppmv; visible light ($\lambda > 420$ nm) intensity of 2.2 mW/cm^2 ; sample amount of 50 mg (with 13 mg of extra water in the case of WO_3 slurry and $\text{NaIO}_4/\text{WO}_3$ slurry); reaction temperature of 30 $^\circ\text{C}$.

Response to Reviewer #2:

This manuscript presents an interesting study on the role of a periodic acid (PA) layer on photocatalytic materials such as WO_3 to enhance the photocatalytic degradation performance for hydrophilic volatile organic contaminants. PA provides hygroscopic properties resulting in air-water-catalyst triphasic system. The PA layer also serves as an effective electron acceptor that results in higher apparent photonic efficiencies. The authors determined apparent photonic efficiencies for degradation of acetaldehyde and some other hydrophilic organics that are obviously larger than those in the absence of PA layer. I believe the authors present an interesting concept and convincing data. There are some aspects that need to be considered and addressed for a practical application of such process. Some comments are below.

Comment 1. Related to lines 82-84. What is the practicality of this for indoor air environment? Will this PA stay there for a long time to perform such function? Have the authors simulated this in a long-term operation performance test? In the field of indoor air quality, what could foul this layer? Will the presence of siloxanes be an issue?

Response: Thank you for valuable comments. To address the durability of PA component, an additional experiment (Supplementary Figure 6) and the following sentences were added:

(p. 6) “To check the long term durability of the PA component, the PA/WO_3 sample that had been stored for six months under ambient condition was tested for its PCD activity, which was little different from that of the fresh PA/WO_3 . This demonstrates that the PA/WO_3 sample can be kept in long-term storage without losing its catalytic activity (*see* Supplementary Figure 6).”

(Supplementary Figure 6) “The PCD activity of PA/WO_3 stored under ambient conditions for six months (*denoted as PA/WO₃-stored*) was tested after drying at 80 °C. It was found that the PA/WO_3 -stored still has good hygroscopicity and its photocatalytic activity is almost the same with that of fresh PA/WO_3 , indicating that the sample remained stable in the long-term storage.”

Supplementary Figure 6. PCD activities of fresh PA/WO_3 and PA/WO_3 stored for six months. The dashed lines with the open symbols represent the CO_2 generated from AA degradation. Experimental conditions: $[\text{AA}]_0 = 120$ ppmv; visible light ($\lambda > 420$ nm) intensity of 2.2 mW/cm²; sample amount of 50 mg; RH 65%; reaction temperature of 30 °C.

To discuss the practicality of the PA/WO₃ system, the following sentences were added:

(p. 15-16) “The multiple roles of PA induce a water layer on the photocatalyst surface, dissolve and concentrate hydrophilic VOCs in the water layer, and make more holes available to produce OH radicals. As a result, PA/WO₃ photocatalyst is even more active than Pt-loaded WO₃ (a popular but expensive visible-light active photocatalyst) for the degradation of acetaldehyde.

PA is proposed as a low-cost component to replace costly Pt cocatalyst for the removal of hydrophilic VOCs. PA is nonvolatile and comparatively nontoxic and its release into the air can be safely neglected. The uptake of water vapor onto PA/WO₃ and the subsequent drying are reversible. The gradual depletion of periodate is the critical problem that should be solved for practical applications. A possible solution is to employ PA/WO₃ as a replaceable filter component in air purifiers. The used filter can be regenerated by recycling iodate back to periodate *via* electrochemical⁶⁰ or other economical chemical oxidation methods. Alternatively, a photoelectrochemical (PEC) filter device that can regenerate *in-situ* periodate as soon as it is converted into iodate under irradiation in the PA/WO₃ filter plans to be developed in a further study. Therefore, for practical applications, PA/WO₃ should be more suitable to be employed as a replaceable filter component, rather than to be coated over the support with a large surface area. On the other hand, hydrophobic VOCs (e.g., DCM, C5, ClC3, and Tol) were little degraded on PA/WO₃ in the same PCD experimental conditions due to their poor solubility in the *in-situ* water layer. Although the inability of PA/WO₃ to remove hydrophobic VOCs is a serious limitation for general purpose applications, its ability to degrade hydrophilic VOCs selectively against hydrophobic VOCs can be exploited in a specific application (*see* Figure 6). In addition, the *in-situ* formed water layer can protect the photocatalyst surface from fouling with inhibiting the deposition of hydrophobic components (*e.g.*, indoor particulate matters like cooking particles, various VOCs containing halogen/phosphorus/silicon found in indoor environments)⁶¹⁻⁶³. As a viable strategy for exploiting *in-situ* water layer formation in ambient air photocatalysis, the PA/photocatalyst may be combined in a hybrid air treatment system where hydrophilic VOCs (that can be rapidly degraded by photocatalysis) are selectively degraded by the PA/photocatalyst and then hydrophobic VOCs (that are more recalcitrant against photocatalysis) are removed by other methods such as adsorption and thermal catalysis. Such approach combines the advantages of various technologies to develop more efficient and economical method of controlling VOCs.”

Comment 2. Related to lines L131-136. While the authors somewhat discuss this, I feel more discussion is needed on the role of surface acidity. What is the role of acidity in this layer? How PA changes the surface charge of the respective photocatalysts? Do authors have quantitative analysis on this aspect?

Response: An additional test of the acidity effect was carried out (*see* Supplementary Figure 9) and the following sentences were added to discuss this point:

(p. 9) “The PA acidity in PA/WO₃ may play a role as well since the pH of the PA solution was 1.5 whereas that of NaIO₄ solution was 4.6. To test the acidity effect, the pH of NaIO₄ solution was adjusted

to 1.5 with adding iodic acid when preparing NaIO₄/WO₃ and the PCD activity of the *acidified* NaIO₄/WO₃ slurry was much higher than that of NaIO₄/WO₃ slurry (see Supplementary Figure 9). Considering that iodic acid alone did not promote the PCD activity of WO₃, the above result implies that the PA acidity in the water layer should contribute to the high PCD activity of PA/WO₃. This might be related with the fact that the reduction of periodate is favored at an acidic condition (IO₄⁻ + 2H⁺ + 2e⁻ → IO₃⁻ + H₂O).”

Supplementary Figure 9. PCD activities of (I) PA/WO₃, (II) acidified NaIO₄/WO₃ slurry (iodic acid was used to adjust pH to 1.5), (III) NaIO₄/WO₃ slurry, (IV) HIO₃/WO₃ slurry and (V) WO₃ slurry. The dashed lines with open symbols represent the CO₂ generated from AA degradation. Experimental conditions: all slurry samples containing 25 mg WO₃ and 13 mg water; [AA]₀ = 120 ppmv; visible light (λ > 420 nm) intensity of 2.2 mW/cm²; RH 65%; reaction temperature of 30 °C.

Comment 3. Related to line L147-148. Relevant to point 1 above, how this will be relevant if you use this PA layer in a real indoor air application? Will you need to coat all indoor air surfaces?

Response: The following sentences were added to discuss the practical aspects of applications.
 (p. 16) “PA is proposed as a low-cost component to replace costly Pt cocatalyst for the removal of hydrophilic VOCs. PA is nonvolatile and comparatively nontoxic and its release into the air can be safely neglected. The uptake of water vapor onto PA/WO₃ and the subsequent drying are reversible. The gradual depletion of periodate is the critical problem that should be solved for practical applications. A possible solution is to employ PA/WO₃ as a replaceable filter component in air purifiers. The used filter can be regenerated by recycling iodate back to periodate *via* electrochemical⁶⁰ or other economical chemical oxidation methods. Alternatively, a photoelectrochemical (PEC) filter device that can regenerate *in-situ* periodate as soon as it is converted into iodate under irradiation in the PA/WO₃ filter plans to be developed in a further study. Therefore, for practical applications, PA/WO₃ should be more suitable to be employed as a replaceable filter component, rather than to be coated over the support with a large surface area.”

Comment 4. Related to line L161. Do authors have more detailed insights through thermodynamic analysis with respect to condensation of water?

Response: To address this issue, the following sentences were added:

(p. 7) “The strong hydrogen bonding between the hydroxyl groups of PA and water molecules should make the condensation of water vapor highly exothermic at ambient condition, where the negative ΔH outweighs the entropy decrease of water vapor condensation to make the overall condensation process thermodynamically spontaneous ($\Delta G < 0$).”

Comment 5. Related to lines L221-223. On the formation of surface charge transfer complex, can authors provide theoretical DFT calculations to examine further this effect?

Response: To address this question, the following sentences were added:

(p. 11) “Moreover, the DFT calculation that was carried out to investigate the interaction between PA and WO_3 surface shows that the calculated adsorption/binding energy of PA on WO_3 surface is -3.7 eV, a high value which usually implies the formation of strong chemical bonds⁵³. The charge density difference analysis shows that there are an electron-depleted region on the WO_3 surface and an electron-gaining region around the iodine center. This clearly indicates that the charge is transferred from the WO_3 surface to PA, which is consistent with the XPS results (*see* Supplementary Figure 12).”

(p. 20-21) “**Computational details**

Density function theory (DFT) calculations were performed by using the CP2K package⁶⁵. PBE functional⁶⁶ with Grimme D3 correction⁶⁷ was used to describe the system. Unrestricted Kohn-Sham DFT has been used as the electronic structure method in the framework of the Gaussian and plane waves method⁶⁸. The Goedecker-Teter-Hutter (GTH) pseudopotentials⁶⁹, DZVPMOLOPT-GTH basis sets⁶⁸ were utilized to describe the molecules. A plane-wave energy cut-off of 500 Ry has been employed.

The charge density difference is defined as

$$\Delta\rho = \rho_{\text{mol/sur}} - \rho_{\text{mol}} - \rho_{\text{sur}}$$

where $\rho_{\text{mol/sur}}$, ρ_{mol} and ρ_{sur} are the electron density of the molecule adsorbed on surface, the molecule, and the surface, respectively. The binding energy is defined as

$$E_b = E_{\text{mol/sur}} - E_{\text{mol}} - E_{\text{sur}}$$

where $E_{\text{mol/sur}}$, E_{mol} and E_{sur} are the energy of the molecule adsorbed on surface, the molecule, and the surface, respectively.”

Supplementary Figure 12. Charge density difference in PA/WO₃ system: (A) the 3-dimensional iso-density of charge density difference. Color code: tungsten (gray), oxygen (red), iodine (purple), hydrogen (white). Isosurface: yellow color indicates the gaining of electron density while cyan color indicates the losing of electron density; (B) the x–y plane averaged charge difference with charge density (ρ) on the x-axis and the distance (Å) in z-direction of the interface unit cell on the y-axis.

Comment 6. Related to lines L238-247; 251-252, Fig. 4C and other relevant sections. Can authors describe how they plan to address the issue of reduced efficiency with time. Such coating should be effective for long term performance. If activity is reduced over time, how will the authors use them in a practical long-term indoor air application? So, how can authors solve the problem of reactivity of PA?

Response: We agree to the reviewer’s point and fully understand the limitations of the studied system. Even though it does not have ideal characteristics for general applications, its unique property should have a practical value for specific applications. To discuss the practical aspects of applications, the following sentences were added:

(p. 16) “PA is proposed as a low-cost component to replace costly Pt cocatalyst for the removal of hydrophilic VOCs. PA is nonvolatile and comparatively nontoxic and its release into the air can be safely neglected. The uptake of water vapor onto PA/WO₃ and the subsequent drying are reversible. The gradual depletion of periodate is the critical problem that should be solved for practical applications. A possible solution is to employ PA/WO₃ as a replaceable filter component in air purifiers. The used filter can be regenerated by recycling iodate back to periodate *via* electrochemical⁶⁰ or other economical chemical oxidation methods. Alternatively, a photoelectrochemical (PEC) filter device that can regenerate *in-situ* periodate as soon as it is converted into iodate under irradiation in the PA/WO₃ filter plans to be developed in a further study. Therefore, for practical applications, PA/WO₃ should be more suitable to be employed as a replaceable filter component, rather than to be coated over the support with a large surface area.”

Comment 7. Supplementary Table 3. Can authors compare the AQE of their WO₃ with those of other such catalysts? Can authors make sure their efficiencies are not overestimated? For example these values reported in Fig. 4 appear higher than values reported in this Table for similar concentration of AA.

Response: To address this question, the AQE calculations were rechecked and the Supplementary Table 4 that compares the WO₃-based photocatalysts for typical VOCs degradation was added. In addition, the following parts were revised to describe the AQE calculation more clearly:

(p. 5) “In addition, the AQE of PA/WO₃ is also outstanding to our knowledge among the reported WO₃-based visible-light PCD systems (Supplementary Table 4).”

(p. 21-22) “For the measurement of AQE, two monochromatic LEDs (Luna Fiber Optic Korea, 2.0 mW/cm²) which emit light at ca. 365 nm and 460 nm were used in the PCD of AA. Except for the light source, all the conditions were the same as that in the PCD tests using the mercury lamp. The AQE of AA degradation was indirectly calculated based on the CO₂ generation rate within 40 min PCD reaction to rule out the effect of AA adsorption on the catalyst surface and reactor surface. The photocatalytic degradation of AA can be divided into the following two half-reactions.

The full degradation of an AA molecule leads to the production of two CO₂ molecules, which is a 10-electron oxidation. Therefore, the production of one CO₂ molecule needs a 5-electron oxidation, which should consume 5 photons: the AQE estimated from the CO₂ production measurement is based on this stoichiometry. The number of incident photons indicates the total number of photons reaching the surface of the catalyst during the reaction time, which were calculated by Eq. (1). The number of reacted photons indicates the number of photons that are utilized in transforming AA molecules into CO₂, which was calculated by Eq. (2). The AQE was calculated by Eq. (2)/Eq. (1). Such estimation of AQE is based on the assumption that AA molecules are fully mineralized to CO₂ with negligible generation of intermediates. As the PCD of AA on PA/WO₃ accompanied the stoichiometric production of CO₂ (see Figures 1, 2, 4), the formation of intermediates (*if any*) should be negligible in terms of the carbon mass balance.

$$\text{Number of incident photons} = \frac{E\lambda At}{hc} \quad (1)$$

$$\text{Number of reacted photons} = 5R_{\text{CO}_2}N_A t \quad (2)$$

$$\text{AQE} = \frac{\text{Number of reacted photons}}{\text{Number of incident photons}} \times 100\% \quad (3)$$

where, E: intensity of LED (mW/cm²); λ: wavelength of LED; A: irradiation area of photocatalysts; t: illumination/reaction time (40 min); h: Planck constant, 6.626 × 10⁻³⁴ Js; c: velocity of light; R_{CO₂}: CO₂ generation rate (mol/min); N_A: Avogadro constant, 6.02 × 10²³/mol.”

Supplementary Table 4. WO₃-based photocatalysts for typical VOCs degradation under visible light

Catalyst type	Catalyst amount /Reactor	VOC type/concentration	Light source	AQE	Ref.
WO ₃ /TiO ₂ nanotubes	2 cm ² film/15 mL (b) [#]	Isopropanol/165 ppmv	LED: λ = 400 nm, I = 112 mW/cm ²	0.35%	[12]
CuO/WO ₃	150 mg/4.4 ml (b)	Acetic acid/ Evaporation: 2 μL liquid acetic acid	29 μW xenon lamp with monochromator: λ = 400 nm	6.3%	[2]
CuO/WO ₃	150 mg/4.4 ml (b)	Formaldehyde/ Evaporation: 2 μL 16 wt % formaldehyde solution	29 μW xenon lamp with monochromator: λ = 400 nm	2.3%	[2]
WC/WO ₃	-/- (b)	Isopropanol/300 ppmv	Xe lamp with UV-cutoff filter: λ = 400 -530 nm I = 0.67 mW/cm ²	3.2%	[13]
Pt/WO ₃	Suspension catalyst (5 g/L) /330 mL (b)	Acetic acid/-	300 W xenon lamp with monochromator: λ = 400 nm	~10%	[14]
PA/WO₃	50 mg/300 mL (b)	120 ppmv	LED: λ = 460 nm, I = 2.0 mW/cm²	16.1 %	This work
PA/WO₃	50 mg/300 mL (b)	700 ppmv	LED: λ = 460 nm, I = 2.0 mW/cm²	64.3%	This work

[#](b) stands for batch reactor.

Response to Reviewer #3:

In this manuscript, the authors developed a novel strategy to introduce a periodic acid (PA) layer on WO_3 to dramatically enhance the photocatalytic activity for air purifying. They found that water vapor in ambient air could be condensed on the surface of WO_3 to form air/water/ WO_3 structure. By this method, a high apparent quantum efficiency of 64.3% at 460 nm for acetaldehyde degradation was obtained. Further study suggested that periodic acid plays multiple roles as a water-layer inducer, a surface complexing ligand that enhances visible light absorption, and an electron acceptor. Moreover, other hydrophilic VOCs (methanol, isopropanol, and acetone) were also rapidly degraded by PA/ WO_3 , but hydrophobic VOCs were not. This study reports some new and interesting results and the presentation is also clear. However, some concerns should be addressed before this manuscript can be considered for publication.

Comment 1. PA is used as an electron acceptor due to its strong oxidation, which can be consumed during the photocatalytic reaction (see Figure 4). Strictly speaking, PA/ WO_3 is not a photocatalyst. Except for PA, is it possible that WO_3 also changes after the reaction? More characterization (TEM, SEM and XPS) of the composite samples before and after the reaction should be carried out. Moreover, PA is toxic, why did the authors use it for air purifying?

Response: Additional analyses of TEM, FE-SEM, and XPS were carried out and compared between the fresh and used WO_3 samples (*see* Supplementary Figure 11 and Supplementary Figure 13) and the following sentence was added:

(p. 12) “FE-SEM, TEM and XPS results showed that the morphology and surface chemical state of WO_3 in PA/ WO_3 were little changed after the PCD reaction of AA (*see* Supplementary Figure 11, Supplementary Figure 13).”

(Supplementary Figure 11) “The morphology of WO_3 -treated and WO_3 -untreated were observed by FE-SEM and TEM. The morphology of untreated WO_3 was irregular nanoparticles with variable sizes ranging from 20 to 100 nm. No obvious difference was observed in the FE-SEM and TEM image of the treated WO_3 , indicating that the PA treatment processing negligibly affected the size and morphology of WO_3 . The possible morphology changes of WO_3 after the PCD of AA was also investigated using FE-SEM and TEM. To eliminate the interference of abundant PA to the results, the PA/ WO_3 used after PCD of AA was washed thoroughly with water (*denoted as WO_3 -used*). As shown above, no obvious change was found in the morphology of WO_3 -used compared with that of fresh WO_3 (namely WO_3 -untreated), indicating that the size and morphology of WO_3 were little changed during the PCD reaction of AA.”

Supplementary Figure 11. (A) FE-SEM and (B) TEM image of (a) WO_3 -untreated, (b) WO_3 -treated and (c) WO_3 -used.

Supplementary Figure 13. W 4f XPS spectra of the fresh and used PA/ WO_3 samples.

To discuss the practical issue regarding the use of PA, the following sentences were added.

(p. 16) “PA is proposed as a low-cost component to replace costly Pt cocatalyst for the removal of hydrophilic VOCs. PA is nonvolatile and comparatively nontoxic and its release into the air can be safely neglected. The uptake of water vapor onto PA/ WO_3 and the subsequent drying are reversible. The gradual depletion of periodate is the critical problem that should be solved for practical applications. A possible solution is to employ PA/ WO_3 as a replaceable filter component in air purifiers. The used filter can be regenerated by recycling iodate back to periodate *via* electrochemical⁶⁰ or other economical chemical oxidation methods. Alternatively, a photoelectrochemical (PEC) filter device that can regenerate *in-situ* periodate as soon as it is converted into iodate under irradiation in the PA/ WO_3 filter

plans to be developed in a further study. Therefore, for practical applications, PA/WO₃ should be more suitable to be employed as a replaceable filter component, rather than to be coated over the support with a large surface area.”

Comment 2. Isotope experiments should be carried out to confirm CO₂ products from VOC degradation, not from some contaminants.

Response: We could not carry out the isotope experiments because we could not obtain the isotopically labeled standard gas of VOCs (mainly acetaldehyde in this study). However, it is very clear that CO₂ should be produced from the target VOCs in this work for the following reasons, which were added in the revised manuscript.

(1) “To test the air tightness of the reactor, it was filled with high-purity synthetic air (79% N₂/21% O₂) and exposed to ambient air. A negligible increase in CO₂ (< 4 ppmv within 60 min) was detected, indicating good air tightness of the reactor system.” (p. 18)

(2) “Before each PCD experiment, the glass reactor was purged with high-purity synthetic air (79% N₂/21% O₂) and irradiated under the mercury lamp or LED lamp to degrade any organic impurities remaining on the photocatalyst surface. The cleaning irradiation continued until the photogeneration of CO₂ was not detected.” (p. 18)

Comment 3. More calculation details on apparent quantum efficiency should be offered. For example, what does the number of reacted photons mean? how much are the t₁ and t₂, respectively. All of the oxidation products should be measured to calculate apparent quantum efficiency. Different from water splitting, VOC degradation reaction is a chain reaction, apparent quantum efficiency is possibly much higher than 100%. The authors should give some comments on so-called high apparent quantum efficiency in this study.

Response: The AQE calculation description is revised to explain it in clearer details as shown below. (p. 21-22) For the measurement of AQE, two monochromatic LEDs (Luna Fiber Optic Korea, 2.0 mW/cm²) which emit light at ca. 365 nm and 460 nm were used in the PCD of AA. Except for the light source, all the conditions were the same as that in the PCD tests using the mercury lamp. The AQE of AA degradation was indirectly calculated based on the CO₂ generation rate within 40 min PCD reaction to rule out the effect of AA adsorption on the catalyst surface and reactor surface. The photocatalytic degradation of AA can be divided into the following two half-reactions.

The full degradation of an AA molecule leads to the production of two CO₂ molecules, which is a 10-electron oxidation. Therefore, the production of one CO₂ molecule needs a 5-electron oxidation, which should consume 5 photons: the AQE estimated from the CO₂ production measurement is based on this stoichiometry. The number of incident photons indicates the total number of photons reaching the

surface of the catalyst during the reaction time, which were calculated by Eq. (1). The number of reacted photons indicates the number of photons that are utilized in transforming AA molecules into CO₂, which was calculated by Eq. (2). The AQE was calculated by Eq. (2)/Eq. (1). Such estimation of AQE is based on the assumption that AA molecules are fully mineralized to CO₂ with negligible generation of intermediates. As the PCD of AA on PA/WO₃ accompanied the stoichiometric production of CO₂ (see Figures 1, 2, 4), the formation of intermediates (if any) should be negligible in terms of the carbon mass balance.

$$\text{Number of incident photons} = \frac{E\lambda At}{hc} \quad (1)$$

$$\text{Number of reacted photons} = 5R_{\text{CO}_2}N_{\text{A}}t \quad (2)$$

$$\text{AQE} = \frac{\text{Number of reacted photons}}{\text{Number of incident photons}} \times 100\% \quad (3)$$

where, E: intensity of LED (mW/cm²); λ: wavelength of LED; A: irradiation area of photocatalysts; t: illumination/reaction time (40 min); h: Planck constant, 6.626 × 10⁻³⁴ Js; c: velocity of light; R_{CO₂}: CO₂ generation rate (mol/min); N_A: Avogadro constant, 6.02 × 10²³/mol.

To further discuss the high apparent quantum efficiency of PA/WO₃ in this study, the Supplementary Table 4 that compares the WO₃-based photocatalysts for typical VOCs degradation was added and the following paragraph was added in p.17-18 of Supplementary information.

“In general, it is difficult to make a direct comparison of the performance of photocatalysts, because the efficiency depends on the experimental conditions such as light intensity, reactor type, and substrate concentration. AQE is usually a good indicator to compare the activity of photocatalysts because it can normalize different light intensity conditions. Nevertheless, AQE is still affected by the type of pollutants and light conditions. For example, Arai et al. found that the AQEs in the PCD of acetic acid, acetaldehyde, formaldehyde on CuO/WO₃ were significantly different (*J. Phys. Chem. C* 2009, 113, 6602). Therefore, the AQEs obtained in this work are compared with some representative visible-light driven PCD systems for AA degradation in the literature (Supplementary Table 3). The AQE of PA/WO₃ (64.3%) is the highest value ever reported in the PCD of AA under visible light irradiation. Besides, the PCD activities of some typical WO₃-based catalysts for VOCs other than AA are also listed in Supplementary Table 4. PA/WO₃ is also outstanding among these visible-light driven WO₃ PCD systems. This fully demonstrates the advantages of the reported PA/WO₃ as a triphase gas-liquid-solid photocatalytic system over the traditional two-phase gas-solid photocatalytic system in treating AA. The superior photocatalytic activity can be attributed to the enrichment of hydrophilic VOCs in the *in-situ* surface water layer, and the efficient electron scavenging by PA. These findings were not unique to WO₃, and other PA-coated semiconductors (e.g., N-TiO₂, BiVO₄) also showed high performance. This proposes a low-cost and facile way to efficiently eliminate hydrophilic VOCs under visible light.”

Comment 4. From Figure 2 b, it is very difficult to obtain the concentration of H₂O or OH from the peak intensity of FT-IR. Other characterization should be used.

Response: To address this question, the *in-situ* DRIFT tests were additionally carried out (see Figure 2D, 2E) and the following sentences were added:

(p. 8-9) “The relation between AA and the surface water layer was further investigated by *in situ* DRIFT spectroscopy under dark conditions. When exposing PA/WO₃ to 300 ppmv AA/dry air flow (see Figure 2D), no peaks corresponding to ν -OH (3600–3200 cm⁻¹) and ν -H₂O (~1630 cm⁻¹) were found, which indicates that the formation of surface water layer is negligible. The small peak located at 1736 cm⁻¹ (ν -C=O vibration mode in aldehydes) is assigned to AA adsorbed *via* hydrogen bonding with a surface OH group⁴³. On the other hand, the ν -OH (3600–3200 cm⁻¹), ν -H₂O (~1630 cm⁻¹), and ν -C=O (1733 cm⁻¹) peaks all increased significantly when exposing PA/WO₃ to 300 ppmv AA/65% RH air flow, which indicates the formation of surface water layer and the enrichment of AA (see Figure 2E). Moreover, a distinct new peak corresponding to C-O stretching vibration (ν -C-O) for carboxylic acids (1325 cm⁻¹)^{44,45} gradually appeared with time. This implies that the *in-situ* water layer formation facilitates not only adsorption/dissolution of AA but also partial pre-oxidation of AA, which subsequently accelerates the PCD process under irradiation.”

(p. 20) “*In situ* DRIFTS was performed using a FT-IR spectrometer (PerkinElmer, USA) equipped with a diffuse-reflectance cell (PIKE) with a ZnSe window. The catalyst sample was placed in the cell and pretreated at 100 °C to eliminate the effects of adsorbed water. AA of 300 ppmv was introduced into the cell through diluting the standard gas (1000 ppmv AA in N₂) with air. The relative humidity (RH) was adjusted by bubbling the air through a stainless-steel bottle containing deionized water. *In situ* DRIFTS spectra were collected after exposing the samples to the flowing stream for 10 s, 1 min, 3 min, and 5 min, respectively.”

Figure 2. (A) Catalyst weight increase of PA/WO₃, NaIO₄/WO₃, and bare WO₃ samples after exposing each catalyst under humid air (RH 65%) for 30 min; (B) FT-IR spectra of bare WO₃ (a) before and (b) after exposure to humid air (RH 65%) for 30 min; (C) C/C₀ and [CO₂] versus Time (min) for PA/WO₃ under humid air (RH 65%) and dry air (RH 0%) conditions; (D) FT-IR spectra of PA/WO₃ (a) before and (b) after exposure to humid air (RH 65%) for 30 min; (E) FT-IR spectra of PA/WO₃ (a) before and (b) after exposure to humid air (RH 65%) for 30 min; (F) C/C₀ and [CO₂] versus Time (min) for PA/WO₃, NaIO₄/WO₃, and WO₃ under humid air (RH 65%) and dry air (RH 0%) conditions.

after exposing to humid air, and those of PA/WO₃ (c) before and (d) after exposing to humid air for 30 min; (C) PCD activity of PA/WO₃ for acetaldehyde (AA) degradation after exposing to dry and humid air for 30 min; *In situ* DRIFT spectra for (D) PA/WO₃ after exposing the sample to 300 ppmv AA/dry air stream and (E) PA/WO₃ in 300 ppmv AA/RH 65% air stream for (a) 10 s, (b) 1 min, (c) 3 min, and (d) 5 min. The spectra of the dry PA/WO₃ surface were collected and used as the background; (F) PCD activity of bare WO₃, NaIO₄/WO₃, and their slurries containing 26 wt% water for the degradation of AA. The dashed lines with open symbols represent CO₂ concentration generated from AA degradation. Experimental conditions: [AA]₀ = 120 ppmv; visible light ($\lambda > 420$ nm) intensity of 2.2 mW/cm²; sample amount of 50 mg (with 13 mg of extra water in the case of WO₃ slurry and NaIO₄/WO₃ slurry); reaction temperature of 30 °C.

Comment 5. In Figure 3b, is it possible that the reflectance spectra difference between WO₃ and PA/WO₃ comes from different aggregation states of WO₃ in different samples since the absorption edge does not change.

Response: We believe that the reflectance spectra difference between WO₃ and PA/WO₃ should be due to the surface “W-O-I-(OH)_n” complex formation for the following reasons:

- (1) XRD, FE-SEM, TEM and N₂ absorption-desorption results showed that the original crystal form, BET surface area, pore structure and morphology of WO₃ are almost unchanged after PA treatment processing (*see* Figure 3A, Supplementary Figure 10, Supplementary Figure 11);
- (2) XPS results in Figure 3 showed the formation of “W-O-I-(OH)_n” complex, which was further confirmed by the DFT calculations (Supplementary Figure 12). It has been reported that the charge transfer between the grafted electron-withdrawing organic molecules and the metal oxide surface affected the light absorption properties of photocatalysts (*Energy Environ. Sci.*, 2014, 7, 954-966; *Appl. Catal. B Environ.*, 2017, 202, 642-652; *ChemComm*, 2016, 52, 13507-13510).

REVIEWER COMMENTS

Reviewer #1 (Remarks to the Author):

The authors carefully addressed all comments from the reviewers and the new version is suitable to be accepted as it is.

Reviewer #2 (Remarks to the Author):

Review Comments for NCOMMS-20-50917A-revised

" Self-wetting triphase photocatalysis for highly selective removal of hydrophilic 2 volatile organic compounds in air"

The authors have addressed my previous comments, but new comments arise with respect to the intended application to use this as a photocatalytic filter and proposed methods to regenerate periodic acid (PA) or replace the filter. In general, the paper presents interesting aspects on the physicochemical properties and interactions of contaminants and the catalyst, but the authors do not provide a convincing case of the potential of this technology for real applications. More quantitative analysis is needed for how such a technology can be used in practice in a sustainable manner. The following questions need to be addressed.

- 1) Authors need to discuss and do some calculations of how much catalyst is needed to treat a certain amount of air contaminated with key contaminants of interest.
- 2) During the treatment using the filter unit and amount of catalyst + PA, will there be reaction intermediates, including some potentially toxic, that will make it back to the room?
- 3) How such proposed technology can compete with other established indoor air purification systems?
- 4) In the video, please check the editorial quality of the legends in the steps since there are some editorial mistakes.

Reviewer #3 (Remarks to the Author):

In this manuscript, the author described the photocatalytic VOC degradation by WO₃/PA. They emphasized the self-wetting properties of PA on the photocatalytic activities. It was somehow interesting. However, practically, it is not sufficiently important and practical for Nature communications. In general, the photocatalytic VOCs degradation was doubtful because of its efficiency on the categories of pollutants, and on low concentration pollutants. I really doubt on this type of photocatalysts unless they can answer the following 2 questions:

1. The selected VOCs were neither most harmful to the modern society, nor the most difficult chemicals as gaseous pollutants. Comparatively, I am very curious that what is the degradation efficiency of WO₃/PA for alkanes, Benzopyrene, nitrobenzene, formaldehyde, etc.

2. As is well-known that practically, the international gas pollutant standard are generally lower than 0.1 PPMv. For example, the standard of toluene is at around 0.03 PPMv. 300 PPMv, as the author used, is too high for VOCs. And normally photocatalysts are useless with the concentrations of VOCs were a bit higher than the international gas pollutant standard. What is the efficiency when the concentration of VOCs as low as 0.1 PPMv? How long can WO₃/PA can degrade pollutants, e.g. toluene, from 0.1 to lower than 0.03 PPMv in a standard 3 m³ or 30 m³ room?

Response to Reviewer #1:

The authors carefully addressed all comments from the reviewers and the new version is suitable to be accepted as it is.

Response to Reviewer #2:

The authors have addressed my previous comments, but new comments arise with respect to the intended application to use this as a photocatalytic filter and proposed methods to regenerate periodic acid (PA) or replace the filter. In general, the paper presents interesting aspects on the physicochemical properties and interactions of contaminants and the catalyst, but the authors do not provide a convincing case of the potential of this technology for real applications. More quantitative analysis is needed for how such a technology can be used in practice in a sustainable manner. The following questions need to be addressed.

Response: Thank you for raising the practicality issue. In the revised version, we tried to fully address the raised issues point-by-point. In particular, all the previous PCD experiments employed the high concentrations of AA ranging in 120-700 ppmv, which is unrealistically high for indoor environment. To further demonstrate its practical application to a low concentration condition, the PCDs of formaldehyde (FA) on PA/WO₃ were additionally tested at a more realistic low concentration of 500 ppbv, which is now shown in Figure 2. The results confirm that PA/WO₃ has an excellent capacity for degrading hydrophilic VOCs (AA and FA) selectively in wide-range concentrations (500 ppbv–700 ppmv). See the detailed responses below.

Comment 1. Authors need to discuss and do some calculations of how much catalyst is needed to treat a certain amount of air contaminated with key contaminants of interest.

Response: Thank you for valuable comments. The present photocatalytic system consists of two components: PA and WO₃.

(1) The dosage of PA. In theory, 1 mol PA can be used to remove 0.2 mol acetaldehyde (AA). The complete mineralization of one AA molecule ($\text{CH}_3\text{CHO} \rightarrow 2\text{CO}_2 + 10\text{e}^-$) is a 10-electron oxidation process, which should be coupled with the reduction of five periodate molecules to iodate ($5\text{IO}_4^- + 10\text{e}^- \rightarrow 5\text{IO}_3^-$). By measuring the consumption of PA and the degradation of AA, we estimated that: 40.7% of initial PA (25 mg PA in 50 mg PA/WO₃) was reduced to iodate with completely mineralizing 0.29 mg AA during five PCD cycles of AA (*see* Figure 5D). Namely, ~0.15 mol AA was removed with consuming 1 mol PA, which is close to the theoretical value. The following sentences were added to discuss this point:

(p. 18) “The reaction stoichiometry indicates that 5 moles of PA is needed to degrade 1 mole of AA ($\text{CH}_3\text{CHO} + 5\text{IO}_4^- \rightarrow 2\text{CO}_2 + 5\text{IO}_3^-$). The cost of PA replacement should not be a problem because of its low price (99.9% purity, ~10.5 \$/kg in USA). Our analysis showed that 40.7% of initial PA (25 mg PA in 50 mg PA/WO₃) was reduced to iodate along with mineralizing 0.29 mg AA during five PCD cycles of AA (*see* Figure 5D). This corresponds to the consumption of 1 mole PA for the degradation of 0.15

mole AA, which is close to the theoretical value (5:1). Based on this ratio, we estimate that 1 g PA is consumed to purify 317 m³ of indoor air contaminated with [AA] = 90 μg/m³ (10 times higher than the US EPA reference concentration for chronic inhalation exposure⁶⁰, 9 μg/m³).”

(2) The dosage of WO₃. The photocatalytic activities of PA/WO₃ with different mass ratio of PA to WO₃ are shown in Supplementary Figure 4. The following sentences were added to discuss this issue:

(p. 5) “The optimal composition of PA/WO₃ was tested by varying the mixing ratio of PA:WO₃ which exhibits good activity for a wide range of PA:WO₃ mass ratio between 2/3 and 3/2 (see Supplementary Figure 4). The highest PCD activity was observed at the 1:1 mass ratio of PA:WO₃, which was used in the preparation of PA/WO₃.”

Supplementary Figure 4. PCD activity of PA/WO₃ with different PA:WO₃ mass ratio (w/w) for acetaldehyde (AA) degradation. The dashed lines with open symbols represent CO₂ generated from AA degradation. Experimental conditions: [AA]₀ = 120 ppmv; visible light ($\lambda > 420$ nm) intensity of 2.2 mW/cm²; the sample mass (PA+WO₃) fixed at 50 mg for all samples; RH 65%; reaction temperature of 30 °C.

Comment 2. During the treatment using the filter unit and amount of catalyst + PA, will there be reaction intermediates, including some potentially toxic, that will make it back to the room?

Response: Potential gaseous toxic products that can be generated from the PCD of AA may include the following species: (1) degradation intermediates of AA (e.g., formaldehyde, acetic acid, etc.), which were not detected in this PCD system; (2) the release of active iodine species (e.g., I₂, HOI), which can be largely ruled out because only iodate was detected as the reduction product of PA on the catalyst surface (see Figure 5D). The PCD exhaust gas was also analyzed to confirm this. The following sentences were added to address this issue:

(p. 17) “PA is proposed as a low-cost component to replace costly Pt cocatalyst for the removal of hydrophilic VOCs. PA is nonvolatile and comparatively nontoxic and its release into the air can be safely neglected. AA degraded by PA/WO₃ photocatalyst generated no detectable volatile byproducts. To check the possible generation of volatile iodine species (e.g., I₂, HOI) from the transformation of PA during the photocatalytic degradation of AA on PA/WO₃, the treated air was absorbed by phenol solution

(1 mM) and analyzed for iodophenol (a product that should be generated from the reaction with volatile iodine species) by HPLC⁵⁹. No iodophenol was detected, which indicated that no volatile iodine species was present in the treated air. No other gaseous organic products (*e.g.*, formaldehyde, acetic acid) were found during the PCD of AA, which indicates the rapid mineralization of AA. The presence of surface water layer can trap any hydrophilic intermediates and degrade them within the water layer without emitting them into the air phase. This self-wetting tri-phasic (air/water/catalyst) photocatalytic system facilitates the complete degradation of hydrophilic VOCs by providing the *in-situ* water layer where the hydrophilic intermediates/byproducts are more efficiently retained and degraded in the aqueous phase whereas the common biphasic photocatalysis (air/catalyst) often generates gaseous intermediates.”

Comment 3. How such proposed technology can compete with other established indoor air purification systems?

Response: There are various technologies for treating air pollutants, mainly adsorption, thermal-catalytic degradation, biodegradation, et al. Among these technologies, thermal-catalytic degradation is energy intensive, and biodegradation usually requires large-scale facilities, which are not suitable for the treatment of indoor air. Adsorption using activated carbons or highly porous materials is the most commonly practised method for indoor air purification. However, the equilibrium adsorption capacity is significantly reduced at low concentrations (sub-ppm levels) of air pollutants despite the high surface area of the adsorbents. Photocatalytic degradation is widely regarded as an effective technology to remove low concentrations of VOCs, although reports in this area are still limited. To further demonstrate the practical application of the PA/WO₃ photocatalytic system to indoor air purification, the PCD of formaldehyde (FA) on PA/WO₃ was also tested at a more realistic low concentration of 500 ppbv, which is newly added as Figure 2. The following sentences are added to address this additional results:

(p. 3) “A common control method for VOCs is adsorption using porous medium (*e.g.*, activated carbon, zeolite, MOFs, etc.)^{4,5}, but their equilibrium adsorption capacity decreases significantly with lowering VOCs concentration⁶.”

(p. 7) “The above PCD experiments employed the high concentrations of AA ranging in 120-700 ppmv, which is unrealistically high for indoor environment. To demonstrate the performance of the PA/WO₃ photocatalyst in a more realistic condition, the PCD of formaldehyde (FA) on PA/WO₃ was additionally tested at a much lower concentration of 500 ppbv in a larger reactor (1.5 L) (compared with the PCD condition of AA) (*see* Figure 2). FA is a common indoor air pollutant and a human carcinogen classified by the World Health Organization (WHO). The FA PCD tests were conducted under blue LED ($\lambda = 460$ nm) irradiation. After 30 min PCD reaction, the concentration of FA decreased from 500 ppbv to 58 ppbv (lower than the limit concentration allowed by WHO, 80 ppbv³⁸) over 50 mg PA/WO₃. Note that FA could be removed by PCD using as low as 1 mg PA/WO₃. The activity of PA/WO₃ was far higher than bare WO₃ (Figure 2B), and it remained active even after ten PCD cycles (Figure 2C). The above results confirm that PA/WO₃ has an excellent capacity for degrading hydrophilic VOCs (FA and AA) selectively in wide-range concentrations (500 ppbv–700 ppmv). No other existing indoor air purification

methods have such effective, durable and selective capacity for the removal of indoor aldehydes under ambient conditions.”

(p. 22) “The PCD experiments of FA were carried out using a bigger reactor (1.5 L) instead of a 300 mL reactor to demonstrate the photocatalytic air treatment in a larger scale. A 460 nm-emitting LED (Luna Fiber Optic Korea, ICN14D-096) was employed as a light source. Photocatalyst powder (1 mg, 10 mg, 50 mg and 100 mg) was dispersed in water in a quartz glass groove (2 cm × 2 cm), which was dried and placed in the PCD reactor. Before each PCD test, the glass reactor was purged with high-purity synthetic air (79% N₂/21% O₂) and irradiated under LED lamp to clean the photocatalyst surface. FA was introduced into the reactor through a mass flow controller regulating the standard gas (100 ppmv formaldehyde in N₂) flow. The initial concentration of FA was adjusted at 500 ppbv, which is much lower than that of other VOCs (120 ppmv). This represents a more realistic test condition where FA is present in indoor air environment. A photoacoustic gas monitor (LumaSense, INNOVA 1412i) was used to measure the concentrations of FA.”

Figure 2. (A) The PCD of formaldehyde (FA) at 500 ppbv on PA/WO₃. The open symbols represent dark control tests and the filled symbols represent the PCD tests over PA/WO₃ (1:1) with different catalyst mass (e.g., 10 mg catalyst composed of 5 mg PA and 5 mg WO₃); (B) PCD of FA on PA/WO₃, bare WO₃ and PA; (C) Repeated PCD cycles of FA degradation over 50 mg PA/WO₃. Experimental conditions: [FA]₀ = 500 ppbv; blue LED ($\lambda = 460$ nm) intensity of 2.0 mW/cm²; RH 65%; reaction temperature of 30 °C.

Comment 4. In the video, please check the editorial quality of the legends in the steps since there are some editorial mistakes.

Response: Thank you for the careful suggestion. The legends in the video have been carefully checked and corrected to avoid the editorial mistakes.

Response to Reviewer #3:

In this manuscript, the author described the photocatalytic VOC degradation by WO₃/PA. They emphasized the self-wetting properties of PA on the photocatalytic activities. It was somehow interesting. However, practically, it is not sufficiently important and practical for Nature communications. In general, the photocatalytic VOCs degradation was doubtful because of its efficiency on the categories of pollutants, and on low concentration pollutants. I really doubt on this type of photocatalysts unless they can answer the following 2 questions:

Response: Thank you for raising the practicality issue. In the revised version, we tried to fully address the raised issues. In particular, all the previous PCD experiments employed the high concentrations of AA ranging in 120-700 ppmv, which is unrealistically high for indoor environment. To further demonstrate its practical application to a low concentration condition, the PCDs of formaldehyde (FA) on PA/WO₃ were additionally tested at a more realistic low concentration of 500 ppbv, which is now shown in Figure 2. The results confirm that PA/WO₃ has an excellent capacity for degrading hydrophilic VOCs (AA and FA) selectively in widely ranging concentrations (500 ppbv–700 ppmv). Please note that aldehydes (AA and FA) which have been selected as a model test pollutant in this study represent the common indoor air pollutants and have been frequently employed as a test compound for indoor air purification studies. See the detailed responses below.

Comment 1. The selected VOCs were neither most harmful to the modern society, nor the most difficult chemicals as gaseous pollutants. Comparatively, I am very curious that what is the degradation efficiency of WO₃/PA for alkanes, Benzopyrene, nitrobenzene, formaldehyde, etc.

Response: The main aim of this study is to demonstrate the unique photocatalytic behaviors of the PA/WO₃ system that involves the *in-situ* formation of surface water layer and the highly selective degradation of hydrophilic VOCs. To test this photocatalytic behavior, acetaldehyde (AA) was selected as the model substrate, which is frequently employed as the test compound for the photocatalytic degradation of VOCs. After demonstrating the PCD of AA using PA/WO₃, we further tested the PCD behaviors for different kinds of VOCs (especially hydrophilic VOCs and hydrophobic VOCs). In our previous work (*Environ. Sci.: Nano*, 2019, 6, 3185), the published photocatalytic research articles from 1999 to 2018 were classified according to the kind of test VOCs. Based on this survey, some representative VOCs in the main categories were chosen as the target VOCs in the present study: for hydrophilic VOCs, acetaldehyde in aldehydes, methanol and isopropyl alcohol (IPA) in alcohols, acetone in ketones; for hydrophobic VOCs, dichloromethane (DCM) in halogenated hydrocarbons, toluene in aromatics. The above chemicals are used extensively in a variety of industrial and domestic applications and widely present in both indoor and outdoor environments. By testing the above VOCs, we found that the PA/WO₃ photocatalytic system is effective only for the degradation of hydrophilic VOCs and very inert for the degradation of hydrophobic VOCs (see Figure 6). Therefore, it should be inactive for alkanes, benzopyrene, nitrobenzene that the reviewer suggested. However, formaldehyde (FA), another VOC that the reviewer suggested, should be active for the degradation on PA/WO₃ and is an important indoor air pollutant that needs to be tested. Therefore, we tested the PCD of FA at a more

realistic low concentration of 500 ppbv, which is newly shown as Figure 2 and the related discussion and the experimental description are added as follows.

(p. 7) “The above PCD experiments employed the high concentrations of AA ranging in 120-700 ppmv, which is unrealistically high for indoor environment. To demonstrate the performance of the PA/WO₃ photocatalyst in a more realistic condition, the PCD of formaldehyde (FA) on PA/WO₃ was additionally tested at a much lower concentration of 500 ppbv in a larger reactor (1.5 L) (compared with the PCD condition of AA) (*see* Figure 2). FA is a common indoor air pollutant and a human carcinogen classified by the World Health Organization (WHO). The FA PCD tests were conducted under blue LED ($\lambda = 460$ nm) irradiation. After 30 min PCD reaction, the concentration of FA decreased from 500 ppbv to 58 ppbv (lower than the limit concentration allowed by WHO, 80 ppbv³⁸) over 50 mg PA/WO₃. Note that FA could be removed by PCD using as low as 1 mg PA/WO₃. The activity of PA/WO₃ was far higher than bare WO₃ (Figure 2B), and it remained active even after ten PCD cycles (Figure 2C). The above results confirm that PA/WO₃ has an excellent capacity for degrading hydrophilic VOCs (FA and AA) selectively in wide-range concentrations (500 ppbv–700 ppmv). No other existing indoor air purification methods have such effective, durable and selective capacity for the removal of indoor aldehydes under ambient conditions.”

(p. 22) “The PCD experiments of FA were carried out using a bigger reactor (1.5 L) instead of a 300 mL reactor to demonstrate the photocatalytic air treatment in a larger scale. A 460 nm-emitting LED (Luna Fiber Optic Korea, ICN14D-096) was employed as a light source. Photocatalyst powder (1 mg, 10 mg, 50 mg and 100 mg) was dispersed in water in a quartz glass groove (2 cm × 2 cm), which was dried and placed in the PCD reactor. Before each PCD test, the glass reactor was purged with high-purity synthetic air (79% N₂/21% O₂) and irradiated under LED lamp to clean the photocatalyst surface. FA was introduced into the reactor through a mass flow controller regulating the standard gas (100 ppmv formaldehyde in N₂) flow. The initial concentration of FA was adjusted at 500 ppbv, which is much lower than that of other VOCs (120 ppmv). This represents a more realistic test condition where FA is present in indoor air environment. A photoacoustic gas monitor (LumaSense, INNOVA 1412i) was used to measure the concentrations of FA.”

Figure 2. (A) The PCD of formaldehyde (FA) at 500 ppbv on PA/WO₃. The open symbols represent dark control tests and the filled symbols represent the PCD tests over PA/WO₃ (1:1) with different catalyst mass (e.g., 10 mg catalyst composed of 5 mg PA and 5 mg WO₃); (B) PCD of FA on PA/WO₃, bare WO₃ and PA; (C) Repeated PCD cycles of FA degradation over 50 mg PA/WO₃. Experimental conditions: [FA]₀ = 500 ppbv; blue LED ($\lambda = 460$ nm) intensity of 2.0 mW/cm²; RH 65%; reaction temperature of 30 °C.

Comment 2. As is well-known that practically, the international gas pollutant standard are generally lower than 0.1 PPMv. For example, the standard of toluene is at around 0.03 PPMv. 300 PPMv, as the author used, is too high for VOCs. And normally photocatalysts are useless with the concentrations of VOCs were a bit higher than the international gas pollutant standard. What is the efficiency when the concentration of VOCs as low as 0.1 PPMv? How long can WO₃/PA can degrade pollutants, e.g. toluene, from 0.1 to lower than 0.03 PPMv in a standard 3 m³ or 30 m³ room?

Response: Thank you for the valuable comment. We agree with the reviewer that the PCD activity of low concentration VOCs is very important for the practical application of photocatalysts, because the indoor VOCs usually exist in ppbv level concentrations. The present PA/WO₃ system is active for the selective removal of hydrophilic VOCs while the hydrophobic VOCs (e.g., toluene) remain intact during photocatalysis due to the “surface water barrier effect” (see Figure 6). To demonstrate the practicality of the PA/WO₃ system in the low concentration region, we tested the PCD of FA at a more realistic low concentration of 500 ppbv, which is newly shown as Figure 2 and the related discussion and the experimental description are added (see the reply to Comment 1 for Figure 2 and the added texts). And to describe how much catalyst is needed to treat how much of air, the following sentences are added in the discussion section.

(p. 18) “The reaction stoichiometry indicates that 5 moles of PA is needed to degrade 1 mole of AA ($\text{CH}_3\text{CHO} + 5\text{IO}_4^- \rightarrow 2\text{CO}_2 + 5\text{IO}_3^-$). The cost of PA replacement should not be a problem because of its

low price (99.9% purity, ~10.5 \$/kg in USA). Our analysis showed that 40.7% of initial PA (25 mg PA in 50 mg PA/WO₃) was reduced to iodate along with mineralizing 0.29 mg AA during five PCD cycles of AA (*see* Figure 5D). This corresponds to the consumption of 1 mole PA for the degradation of 0.15 mole AA, which is close to the theoretical value (5:1). Based on this ratio, we estimate that 1 g PA is consumed to purify 317 m³ of indoor air contaminated with [AA] = 90 µg/m³ (10 times higher than the US EPA reference concentration for chronic inhalation exposure⁶⁰, 9 µg/m³).”

REVIEWERS' COMMENTS

Reviewer #2 (Remarks to the Author):

The authors addressed most of my comments but I still believe they have not addressed the comment on practicality of this process when it will be used for treating a certain amount of air with multiple contaminants considering contaminated air volume to catalyst surface area ratio, air residence time, and other parameters when implementation of such technology is envisaged for indoor air application. What will be design parameters for the implementation of such technology in practice? These kinds of data and calculations will answer the question if such a technology has real viability for implementation. Authors need to provide more data and address this issue.

Reviewer #3 (Remarks to the Author):

The authors have addressed all comments from the reviewer and the new version is suitable to be accepted as it is.

Response to Reviewer #2:

The authors addressed most of my comments but I still believe they have not addressed the comment on practicality of this process when it will be used for treating a certain amount of air with multiple contaminants considering contaminated air volume to catalyst surface area ratio, air residence time, and other parameters when implementation of such technology is envisaged for indoor air application. What will be design parameters for the implementation of such technology in practice? These kinds of data and calculations will answer the question if such a technology has real viability for implementation. Authors need to provide more data and address this issue.

Response: The practicality of PA/WO₃ application is further discussed with adding a new figure (Supplementary Figure 15) as follows:

(p. 19) “To further investigate the effect of the geometric surface area of the coated photocatalyst, the PCD activity (for removing 500 ppbv FA in 1.5 L air) of PA/WO₃ was compared between the different geometric coating areas of 1 cm² vs. 4 cm² (see Supplementary Figure 15). The PCD activity little decreased with decreasing the catalyst coating area from 4 cm² to 1 cm², which shows that the ratio of treated air volume to catalyst coating area can reach over 1.5 L/cm². This clearly demonstrates that PA/WO₃ has good capacity in purifying large volume of air. In practical applications, this ratio is expected to be further enhanced by optimizing the catalyst dosage and coating thickness. Specific design parameters (e.g., air flow rate, catalyst dosage, thickness and area of catalyst coating, etc) need to be carefully adjusted according to the actual application purpose. These engineering parameters remain to be investigated in future works. It is worth noting that the PA/WO₃ photocatalyst demonstrated good applicability for a wide range of VOC concentrations (500 ppbv–700 ppmv).”

Supplementary Figure 15. The PCD of formaldehyde (FA) (filled symbols) at 500 ppbv over 1 mg PA/WO₃ coated on the area of 4 cm² (2 cm × 2 cm) versus 1 cm² (1 cm × 1 cm). The dark control tests (open symbols) are also compared. Experimental conditions: [FA]₀ = 500 ppbv; reactor volume of 1.5 L; blue LED ($\lambda = 460$ nm) intensity of 2.0 mW/cm²; RH 65%; reaction temperature of 30 °C.

Response to Reviewer #3:

The authors have addressed all comments from the reviewer and the new version is suitable to be accepted as it is.